# Graph Oracle Models, Lower Bounds, and Gaps for Parallel Stochastic Optimization

**Blake Woodworth**
Toyota Technological Institute at Chicago
`blake@ttic.edu`

**Jialei Wang**
Two Sigma Investments
`jialei.wang@twosigma.com`

**Adam Smith**
Boston University
`ads22@bu.edu`

**Brendan McMahan**
Google
`mcmahan@google.com`

**Nathan Srebro**[*]
Toyota Technological Institute at Chicago
`nati@ttic.edu`

## Abstract

We suggest a general oracle-based framework that captures different parallel stochastic optimization settings described by a dependency graph, and derive generic lower bounds in terms of this graph. We then use the framework and derive lower bounds for several specific parallel optimization settings, including delayed updates and parallel processing with intermittent communication. We highlight gaps between lower and upper bounds on the oracle complexity, and cases where the "natural" algorithms are not known to be optimal.

## 1 Introduction

Recently, there has been great interest in stochastic optimization and learning algorithms that leverage parallelism, including e.g. delayed updates arising from pipelining and asynchronous concurrent processing, synchronous single-instruction-multiple-data parallelism, and parallelism across distant devices. With the abundance of parallelization settings and associated algorithms, it is important to precisely formulate the problem, which allows us to ask questions such as "is there a better method for this problem than what we have?" and "what is the best we could possibly expect?"

Oracle models have long been a useful framework for formalizing stochastic optimization and learning problems. In an oracle model, we place limits on the algorithm's access to the optimization objective, but not what it may do with the information it receives. This allows us to obtain sharp lower bounds, which can be used to argue that an algorithm is optimal and to identify gaps between current algorithms and what might be possible. Finding such gaps can be very useful—for example, the gap between the first order optimization lower bound of Nemirovski et al. [21] and the best known algorithms at the time inspired Nesterov's accelerated gradient descent algorithm [22].

We propose an oracle framework for formalizing different parallel optimization problems. We specify the structure of parallel computation using an "oracle graph" which indicates how an algorithm accesses the oracle. Each node in the graph corresponds to a single stochastic oracle query, and that query (e.g. the point at which a gradient is calculated) must be computed using only oracle accesses in ancestors of the node. We generally think of each stochastic oracle access as being based on a single data sample, thus involving one or maybe a small number of vector operations.

In Section 3 we devise generic lower bounds for parallel optimization problems in terms of simple properties of the associated oracle graph, namely the length of the longest dependency chain and the total number of nodes. In Section 4 we study specific parallel optimization settings in which

---

[*]Part of this work was done while visiting Google.

many algorithms have been proposed, formulate them as graph-based oracle parallel optimization problems, instantiate our lower bounds, and compare them with the performance guarantees of specific algorithms. We highlight gaps between the lower bound and the best known upper bound and also situations where we can devise an optimal algorithm that matches the lower bound, but where this is not the "natural" and typical algorithm used in this settings. The latter indicates either a gap in our understanding of the "natural" algorithm or a need to depart from it.

**Previously suggested models**   Previous work studied communication lower bounds for parallel convex optimization where there are $M$ machines each containing a local function (e.g. a collection of samples from a distribution). Each machine can perform computation on its own function, and then periodically every machine is allowed to transmit information to the others. In order to prove meaningful lower bounds based on the number of rounds of communication, it is necessary to prevent the machines from simply transmitting their local function to a central machine, or else any objective could be optimized in one round. There are two established ways of doing this. First, one can allow arbitrary computation on the local machines, but restrict the number of bits that can be transmitted in each round. There is work focusing on specific statistical estimation problems that establishes communication lower bounds via information-theoretic arguments [7, 12, 29]. Alternatively, one can allow the machines to communicate real-valued vectors, but restrict the types of computation they are allowed to perform. For instance, Arjevani and Shamir [3] present communication complexity lower bounds for algorithms which can only compute vectors that lie in a certain subspace, which includes e.g. linear combinations of gradients of their local function. Lee et al. [16] assume a similar restriction, but allow the data defining the local functions to be allocated to the different machines in a strategic manner. Our framework applies to general stochastic optimization problems and does not impose any restrictions on what computation the algorithm may perform, and is thus a more direct generalization of the oracle model of optimization.

Recently, Duchi et al. [10] considered first-order optimization in a special case of our proposed model (the "simple parallelism" graph of Section 4.2), but their bounds apply in a more limited parameter regime, see Section 3 for discussion.

## 2   The graph-based oracle model

We consider the following stochastic optimization problem

$$\min_{x \in \mathbb{R}^m : \|x\| \leq B} F(x) := \mathbb{E}_{z \sim \mathcal{P}} \left[ f(x; z) \right] \tag{1}$$

The problem (1) captures many important tasks, such as supervised learning, in which case $f(x; z)$ is the loss of a model parametrized by $x$ on data instance $z$ and the goal is to minimize the population risk $\mathbb{E}\left[f(x; z)\right]$. We assume that $f(\cdot; z)$ is convex, $L$-Lipschitz, and $H$-smooth for all $z$. We also allow $f$ to be non-smooth, which corresponds to $H = \infty$. A function $g$ is $L$-Lipschitz when $\|g(x) - g(y)\| \leq L \|x - y\|$ for all $x, y$, and it is $H$-smooth when it is differentiable and its gradient is $H$-Lipschitz. We consider optimization algorithms that use either a stochastic gradient or stochastic prox oracle ($\mathcal{O}_{\text{grad}}$ and $\mathcal{O}_{\text{prox}}$ respectively):

$$\mathcal{O}_{\text{grad}}(x, z) = \left(f(x; z), \ \nabla f(x; z)\right) \tag{2}$$

$$\mathcal{O}_{\text{prox}}(x, \beta, z) = \left(f(x; z), \ \nabla f(x; z), \ \text{prox}_{f(\cdot; z)}(x, \beta)\right) \tag{3}$$

$$\text{where} \quad \text{prox}_{f(\cdot; z)}(x, \beta) = \arg\min_y f(y; z) + \frac{\beta}{2} \|y - x\|^2 \tag{4}$$

The prox oracle is quite powerful and provides global rather than local information about $f$. In particular, querying the prox oracle with $\beta = 0$ fully optimizes $f(\cdot; z)$.

As stated, $z$ is an argument to the oracle, however there are two distinct cases. In the "fully stochastic" oracle setting, the algorithm receives an oracle answer corresponding to a random $z \sim \mathcal{P}$. We also consider a setting in which the algorithm is allowed to "actively query" the oracle. In this case, the algorithm may either sample $z \sim \mathcal{P}$ or choose a desired $z$ and receive an oracle answer for that $z$. Our lower bounds hold for either type of oracle. Most optimization algorithms only use the fully stochastic oracle, but some require more powerful active queries.

We capture the structure of a parallel optimization algorithm with a directed, acyclic **oracle graph** $\mathcal{G}$. Its depth, $D$, is the length of the longest directed path, and the size, $N$, is the number of nodes. Each

node in the graph represents a single stochastic oracle access, and the edges in the graph indicate where the results of that oracle access may be used: only the oracle accesses from *ancestors* of each node are available when issuing a new query. These limitations might arise e.g. due to parallel computation delays or the expense of communicating between disparate machines.

Let $\mathcal{Q}$ be the set of possible oracle queries, with the exact form of queries (e.g., $q = x$ vs. $q = (x, \beta, z)$) depending on the context. Formally, a randomized optimization algorithm that accesses the stochastic oracle $\mathcal{O}$ as prescribed by the graph $\mathcal{G}$ is specified by associating with each node $v_t$ a query rule $R_t : (\mathcal{Q}, \mathcal{O}(\mathcal{Q}))^* \times \Xi \to \mathcal{Q}$, plus a single output rule $\hat{X} : (\mathcal{Q}, \mathcal{O}(\mathcal{Q}))^* \times \Xi \to \mathcal{X}$. We grant all of the nodes access to a source of shared randomness $\xi \in \Xi$ (e.g. an infinite stream of random bits). The mapping $R_t$ selects a query $q_t$ to make at node $v_t$ using the set of queries and oracle responses in ancestors of $v_t$, namely

$$q_t = R_t\big((q_i, \mathcal{O}(q_i) \,:\, i \in \text{Ancestors}(v_t)), \xi\big) \tag{5}$$

Similarly, the output rule $\hat{X}$ maps from all of the queries and oracle responses to the algorithm's output as $\hat{x} = \hat{X}((q_i, \mathcal{O}(q_i) : i \in [N]), \xi)$. The essential question is: for a class of optimization problems $(\mathcal{G}, \mathcal{O}, \mathcal{F})$ specified by a dependency graph $\mathcal{G}$, a stochastic oracle $\mathcal{O}$, and a function class $\mathcal{F}$, what is the best possible guarantee on the expected suboptimality of an algorithm's output, i.e.

$$\inf_{\left(R_1, \ldots, R_N, \hat{X}\right)} \sup_{f \in \mathcal{F}} \mathbb{E}_{\hat{x}, z}\left[f(\hat{x}; z)\right] - \min_x \mathbb{E}_z\left[f(x; z)\right] \tag{6}$$

In this paper, we consider optimization problems $(\mathcal{G}, \mathcal{O}, \mathcal{F}_{L,H,B})$ where $\mathcal{F}_{L,H,B}$ is the class of convex, $L$-Lipschitz, and $H$-smooth functions on the domain $\{x \in \mathbb{R}^m : \|x\| \leq B\}$ and parametrized by $z$, and $\mathcal{O}$ is either a stochastic gradient oracle $\mathcal{O}_{\text{grad}}$ (2) or a stochastic prox oracle $\mathcal{O}_{\text{prox}}$ (3). We consider this function class to contain Lipschitz but non-smooth functions too, which corresponds to $H = \infty$. Our function class does not bound the dimension $m$ of the problem, as we seek to understand the best possible guarantees in terms of Lipschitz and smoothness constants that hold in any dimension. Indeed, there are (typically impractical) algorithms such as center-of-mass methods, which might use the dimension in order to significantly reduce the oracle complexity, but at a potentially huge computational cost. Nemirovski [20] studied non-smooth optimization in the case that the dimension is bounded, proving lower bounds in this setting that scale with the $1/3$-power of the dimension but have only logarithmic dependence on the suboptimality. We do not analyze strongly convex functions, but the situation is similar and lower bounds can be established via reduction [28].

## 3 Lower bounds

We now provide lower bounds for optimization problems $(\mathcal{G}, \mathcal{O}_{\text{grad}}, \mathcal{F}_{L,H,B})$ and $(\mathcal{G}, \mathcal{O}_{\text{prox}}, \mathcal{F}_{L,H,B})$ in terms of $L, H, B$, and the depth and size of $\mathcal{G}$.

**Theorem 1.** *Let $L, B \in (0, \infty)$, $H \in [0, \infty]$, $N \geq D \geq 1$, let $\mathcal{G}$ be any oracle graph of depth $D$ and size $N$ and consider the optimization problem $(\mathcal{G}, \mathcal{O}_{grad}, \mathcal{F}_{L,H,B})$. For any randomized algorithm $\mathcal{A} = (R_1, \ldots, R_N, \hat{X})$, there exists a distribution $\mathcal{P}$ and a convex, $L$-Lipschitz, and $H$-smooth function $f$ on a $B$-bounded domain in $\mathbb{R}^m$ for $m = O\left(\max\left\{N^2, D^3 N\right\} \log(DN)\right)$ such that*

$$\mathbb{E}_{\substack{z \sim \mathcal{P} \\ \hat{X} \sim \mathcal{A}}}\left[f(\hat{X}; z)\right] - \min_x \mathbb{E}_{z \sim \mathcal{P}}\left[f(x; z)\right] \geq \Omega\left(\min\left\{\frac{LB}{\sqrt{D}}, \frac{HB^2}{D^2}\right\} + \frac{LB}{\sqrt{N}}\right)$$

**Theorem 2.** *Let $L, B \in (0, \infty)$, $H \in [0, \infty]$, $N \geq D \geq 1$, let $\mathcal{G}$ be any oracle graph of depth $D$ and size $N$ and consider the optimization problem $(\mathcal{G}, \mathcal{O}_{prox}, \mathcal{F}_{L,H,B})$. For any randomized algorithm $\mathcal{A} = (R_1, \ldots, R_N, \hat{X})$, there exists a distribution $\mathcal{P}$ and a convex, $L$-Lipschitz, and $H$-smooth function $f$ on a $B$-bounded domain in $\mathbb{R}^m$ for $m = O\left(\max\left\{N^2, D^3 N\right\} \log(DN)\right)$ such that*

$$\mathbb{E}_{\substack{z \sim \mathcal{P} \\ \hat{X} \sim \mathcal{A}}}\left[f(\hat{X}; z)\right] - \min_x \mathbb{E}_{z \sim \mathcal{P}}\left[f(x; z)\right] \geq \Omega\left(\min\left\{\frac{LB}{D}, \frac{HB^2}{D^2}\right\} + \frac{LB}{\sqrt{N}}\right)$$

These are the tightest possible lower bounds in terms of just the depth and size of $\mathcal{G}$ in the sense that for all $D, N$ there are graphs $\mathcal{G}$ and associated algorithms which match the lower bound. Of course, for specific, mostly degenerate graphs they might not be tight. For instance, our lower bound for the graph consisting of a short sequential chain plus a very large number of disconnected nodes might

be quite loose due to the artificial inflation of $N$. Nevertheless, for many interesting graphs they are tight, as we shall see in Section 4.

Each lower bound has two components: an "optimization" term and a "statistical" term. The statistical term $\Omega(LB/\sqrt{N})$ is well known, although we include a brief proof of this portion of the bound in Appendix D for completeness. The optimization term depends on the depth $D$, and indicates, intuitively, the best suboptimality guarantee that can be achieved by an algorithm using unlimited parallelism but only $D$ rounds of communication. Arjevani and Shamir [3] also obtain lower bounds in terms of rounds of communication, which are similar to how our lower bounds depend on depth. However they restricted the type of computations that are allowed to the algorithm to a specific class of operations, while we only limit the number of oracle queries and the dependency structure between them, but allow forming the queries in any arbitrary way.

Similar to Arjevani and Shamir [3], to establish the optimization term in the lower bounds, we construct functions that require multiple rounds of sequential oracle accesses to optimize. In the gradient oracle case, we use a single, deterministic function which resembles a standard construction for first order optimization lower bounds. For the prox case, we construct two functions inspired by previous lower bounds for round-based and finite sum optimization [3, 28]. In order to account for randomized algorithms that might leave the span of gradients or proxs returned by the oracle, we use a technique that was proposed by Woodworth and Srebro [27, 28] and refined by Carmon et al. [8]. For our specific setting, we must slightly modify existing analysis, which is detailed in Appendix A.

A useful feature of our lower bounds is that they apply when both the Lipschitz constant and smoothness are bounded *concurrently*. Consequently, "non-smooth" in the subsequent discussion can be read as simply identifying the case where the $L$ term achieves the minimum as opposed to the $H$ term (even if $H < \infty$). This is particularly important when studying stochastic parallel optimization, since obtaining non-trivial guarantees in a purely stochastic setting requires some sort of control on the magnitude of the gradients (smoothness by itself is not sufficient), while obtaining parallelization speedups often requires smoothness, and so we would like to ask what is the best that can be done when both Lipschitz and smoothness are controlled. Interestingly, the dependence on both $L$ and $H$ in our bounds is tight, even when the other is constrained, which shows that the optimization term cannot be substantially reduced by using both conditions together.

In the case of the gradient oracle, we "smooth out" a standard non-smooth lower bound construction [21, 27]; previous work has used a similar approach in slightly different settings [2, 13]. For $\ell \leq L$ and $\eta \leq H$, and orthonormal $v_1, \ldots, v_{D+1}$ drawn uniformly at random, we define the $\ell$-Lipschitz but non-smooth function $\tilde{f}$, and its $\ell$-Lipschitz, $\eta$-smooth "$\eta$-Moreau envelope" [5]:

$$\tilde{f}(x) = \max_{1 \leq r \leq D+1} \ell \left( v_r^\top x - \frac{r-1}{2(D+1)^{1.5}} \right) \qquad f(x) = \min_y \tilde{f}(y) + \frac{\eta}{2} \|y - x\|^2 \qquad (7)$$

This defines a distribution over $f$'s based on the randomness in the draw of $v_1, \ldots, v_{D+1}$, and we apply Yao's minimax principle. In Appendix B, we prove Theorem 1 using this construction.

In the case of the prox oracle, we "straighten out" the smooth construction of Woodworth and Srebro [28]. For fixed constants $c, \gamma$, we define the following Lipschitz and smooth scalar function $\phi_c$:

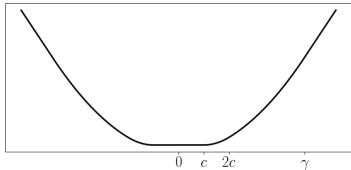

$$\phi_c(z) = \begin{cases} 0 & |z| \leq c \\ 2(|z| - c)^2 & c < |z| \leq 2c \\ z^2 - 2c^2 & 2c < |z| \leq \gamma \\ 2\gamma |z| - \gamma^2 - 2c^2 & |z| > \gamma \end{cases} \qquad (8)$$

For $\mathcal{P} = \text{Uniform}\{1, 2\}$ and orthonormal $v_1, \ldots, v_{2D}$ drawn uniformly at random, we define

$$f(x; 1) = \frac{\eta}{8} \left( -2a v_1^\top x + \phi_c \left( v_{2D}^\top x \right) + \sum_{r=3,5,7,\ldots}^{2D-1} \phi_c \left( v_{r-1}^\top x - v_r^\top x \right) \right) \qquad (9)$$

$$f(x; 2) = \frac{\eta}{8} \left( \sum_{r=2,4,6,\ldots}^{2D} \phi_c \left( v_{r-1}^\top x - v_r^\top x \right) \right) \qquad (10)$$

Again, this defines a distribution over $f$'s based on the randomness in the draw of $v_1, \ldots, v_{2D}$ and we apply Yao's minimax principle. In Appendix C, we prove Theorem 2 using this construction.

| Graph example | With gradient oracle | With gradient and prox oracle |
|---|---|---|
| path($T$) (Section 4.1) | $\frac{L}{\sqrt{T}}$ | |
| layer($T,M$) (Section 4.2) | $\left(\frac{L}{\sqrt{T}}\wedge\frac{H}{T^2}\right)+\frac{L}{\sqrt{MT}}$ | $\left(\frac{L}{T}\wedge\frac{H}{T^2}\right)+\frac{L}{\sqrt{MT}}$ |
| delay($T,\tau$) (Section 4.3) | $\left(\frac{L}{\sqrt{T/\tau}}\wedge\frac{H\tau^2}{T^2}\right)+\frac{L}{\sqrt{T}}$ | $\left(\frac{L\tau}{T}\wedge\frac{H\tau^2}{T^2}\right)+\frac{L}{\sqrt{T}}$ |
| intermittent($T,K,M$) (Section 4.4) | $\left(\frac{L}{\sqrt{KT}}\wedge\frac{H}{K^2T^2}\right)+\frac{L}{\sqrt{MKT}}$ $\frac{L}{\sqrt{KT}}\wedge\left(\frac{H}{T^2}+\frac{L}{\sqrt{MKT}}\right)$ $\wedge\left(\frac{H}{TK}+\frac{L}{\sqrt{MKT}}\right)\log\left(\frac{MKT}{L}\right)$ | $\left(\frac{L}{KT}\wedge\frac{H}{K^2T^2}\right)+\frac{L}{\sqrt{MKT}}$ $\frac{L}{\sqrt{KT}}\wedge\left(\left(\frac{L}{T}\wedge\frac{H}{T^2}\right)+\frac{L}{\sqrt{MKT}}\right)$ $\wedge\left(\frac{H}{TK}+\frac{L}{\sqrt{MKT}}\right)\log\left(\frac{MKT}{L}\right)$ |

Table 1: Summary of upper and lower bounds for stochastic convex optimization of $L$-Lipschitz and $H$-smooth functions with $T$ iterations, $M$ machines, and $K$ sequential steps per machine. Green indicates lower bounds matched only by "unnatural" methods, red and blue indicates a gap between the lower and upper bounds.

**Relation to previous bounds**  As mentioned above, Duchi et al. [10] recently showed a lower bound for first- and zero-order stochastic optimization in the "simple parallelism" graph consisting of $D$ layers, each with $M$ nodes. Their bound [10, Thm 2] applies only when the dimension $m$ is constant, and $D = O(m\log\log M)$. Our lower bound requires non-constant dimension, but applies in any range of $M$. Furthermore, their proof techniques do not obviously extend to prox oracles.

## 4  Specific dependency graphs

We now use our framework to study four specific parallelization structures. The main results (tight complexities and gaps between lower and upper bounds) are summarized in Table 1. For simplicity and without loss of generality, we set $B = 1$, i.e. we normalize the optimization domain to be $\{x \in \mathbb{R}^m : \|x\| \le 1\}$. All stated upper and lower bounds are for the expected suboptimality $\mathbb{E}[F(\hat{x})] - F(x^*)$ of the algorithm's output.

### 4.1  Sequential computation: the path graph 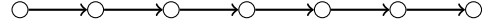

We begin with the simplest model, that of sequential computation captured by the path graph of length $T$ depicted above. The ancestors of each vertex $v_i$, $i = 1\ldots T$ are all the preceding vertices $(v_1,\ldots,v_{i-1})$. The sequential model is of course well studied and understood. To see how it fits into our framework: A path graph of length $T$ has a depth of $D = T$ and size of $N = T$, thus with either gradient or prox oracles, the statistical term is dominant in Theorems 1 and 2. These lower bounds are matched by sequential stochastic gradient descent, yielding a tight complexity of $\Theta(L/\sqrt{T})$ and the familiar conclusion that SGD is (worst case) optimal in this setting.

### 4.2  Simple parallelism: the layer graph 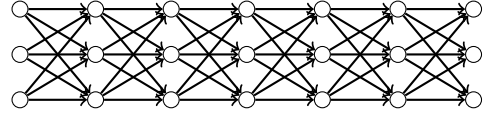

We now turn to a model in which $M$ oracle queries can be made in parallel, and the results are broadcast for use in making the next batch of $M$ queries. This corresponds to synchronized parallelism and fast communication between processors. The model is captured by a layer graph of width $M$, depicted above for $M = 3$. The graph consists of $T$ layers $i = 1,\ldots,T$ each with $M$ nodes $v_{t,1},\ldots,v_{t,m}$ whose ancestors include $v_{t',i}$ for all $t' < t$ and $i \in [M]$. The graph has a depth of $D = T$ and size of $N = MT$. With a stochastic gradient oracle, Theorem 1 yields a lower bound of:

$$\Omega\left(\min\left\{\frac{L}{\sqrt{T}},\frac{H}{T^2}\right\}+\frac{L}{\sqrt{MT}}\right) \tag{11}$$

which is matched by accelerated mini-batch SGD (A-MB-SGD) [9, 15], establishing the optimality of A-MB-SGD in this setting. For sufficiently smooth objectives, the same algorithm is also optimal even if prox access is allowed, since Theorem 2 implies a lower bound of:

$$\Omega\left(\min\left\{\frac{L}{T},\frac{H}{T^2}\right\}+\frac{L}{\sqrt{MT}}\right). \tag{12}$$

That is, for smooth objectives, having access to a prox oracle does not improve the optimal complexity over just using gradient access. However, for non-smooth or insufficiently smooth objectives, there is a gap between (11) and (12). An optimal algorithm, smoothed A-MB-SGD, uses the prox oracle in order to calculate gradients of the Moreau envelope of $f(x; z)$ (cf. Proposition 12.29 of [5]), and then performs A-MB-SGD on the smoothed objectives. This yields a suboptimality guarantee that precisely matches (12), establishing that the lower bound from Theorem 2 is tight for the layer graph, and that smoothed A-MB-SGD is optimal. An analysis of the smoothed A-MB-SGD algorithm is provided in Appendix E.1.

### 4.3 Delayed updates

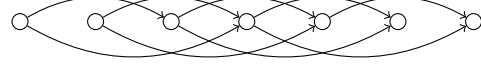

We now turn to a delayed computation model that is typical in many asynchronous parallelization and pipelined computation settings, e.g. when multiple processors or machines are working asynchronously, reading iterates, taking some time to perform the oracle accesses and computation, then communicating the results back (or updating the iterate accordingly) [1, 6, 17, 19, 25]. This is captured by a "delay graph" with $T$ nodes $v_1, \ldots, v_T$ and delays $\tau_t$ for the response to the oracle query performed at $v_t$ to become available. Hence, Ancestors$(v_t) = \{v_s \mid s + \tau_s \leq t\}$. Analysis is typically based on the delays being bounded, i.e. $\tau_t \leq \tau$ for all $t$. The depiction above corresponds to $\tau_t = 2$; the case $\tau_t = 1$ corresponds to the path graph. With constant delays $\tau_t = \tau$, the delay graph has depth $D \leq T/\tau$ and size $N = T$, so Theorem 1 gives the following lower bound when using a gradient oracle:

$$\Omega\left(\min\left\{\frac{L}{\sqrt{T/\tau}}, \frac{H}{(T/\tau)^2}\right\} + \frac{L}{\sqrt{T}}\right). \tag{13}$$

Delayed SGD, with updates $x_t \leftarrow x_{t-1} - \eta_t \nabla f(x_{t-\tau_t}; z)$, is a natural algorithm in this setting. Under the bounded delay assumption the best guarantee we are aware of for delayed update SGD is (see [11] improving over [1])

$$O\left(\frac{H}{T/\tau^2} + \frac{L}{\sqrt{T}}\right). \tag{14}$$

This result is significantly worse than the lower bound (13) and quite disappointing. It does not provide for a $1/T^2$ accelerated optimization rate, but even worse, compared to non-accelerated SGD it suffers a slowdown quadratic in the delay, compared to the linear slowdown we would expect. In particular, the guarantee (14) only allows maximum delay of $\tau = O(T^{1/4})$ in order to attain the optimal statistical rate $\Theta(L/\sqrt{T})$, whereas the lower bound allows a delay up to $\tau = O(T^{3/4})$.

This raises the question of whether a different algorithm can match the lower bound (13). The answer is affirmative, but it requires using an "unnatural" algorithm, which simulates a mini-batch approach in what seems an unnecessarily wasteful way. We refer to this as a "wait-and-collect" approach: it works in $T/(2\tau)$ stages, each stage consisting of $2\tau$ iterations (i.e. nodes or oracle accesses). In stage $i$, $\tau$ iterations are used to obtain $\tau$ stochastic gradient estimates $\nabla f(x_i; z_{2\tau i+j})$, $j = 1, \ldots, \tau$ at the same point $x_i$. For the remaining $\tau$ iterations, we wait for all the preceding oracle computations to become available and do not even use our allowed oracle access. We can then finally update the $x_{i+1}$ using the minibatch of $\tau$ gradient estimates. This approach is also specified formally as Algorithm 2 in Appendix E.2. Using this approach, we can perform $T/(2\tau)$ A-MB-SGD updates with a minibatch size of $\tau$, yielding a suboptimality guarantee that precisely matches the lower bound (13).

Thus (13) indeed represents the tight complexity of the delay graph with a stochastic gradient oracle, and the wait-and-collect approach is optimal. However, this answer is somewhat disappointing and leaves an intriguing open question: can a more natural, and seemingly more efficient (no wasted oracle accesses) delayed update SGD algorithm also match the lower bound? An answer to this question has two parts: first, does the delayed update SGD truly suffer from a $\tau^2$ slowdown as indicated by (14), or does it achieve linear degradation and a speculative guarantee of

$$O\left(\frac{H}{T/\tau} + \frac{L}{\sqrt{T}}\right). \tag{15}$$

Second, can delayed update SGD be accelerated to achieve the optimal rate (13). We note that concurrent with our work there has been progress toward closing this gap: Arjevani et al. [4] showed an improved bound matching the non-accelerated (15) for delayed updates (with a fixed delay) on

quadratic objectives. It still remains to generalize the result to smooth non-quadratic objectives, handle non-constant bounded delays, and accelerate the procedure so as to improve the rate to $(\tau/T)^2$.

## 4.4 Intermittent communication

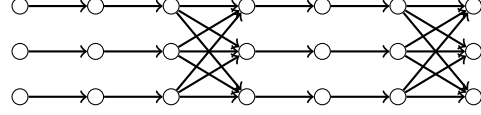

We now turn to a parallel computation model which is relevant especially when parallelizing across disparate machines: in each of $T$ iterations, there are $M$ machines that, instead of just a single oracle access, perform $K$ sequential oracle accesses before broadcasting to all other machines synchronously. This communication pattern is relevant in the realistic scenario where local computation is plentiful relative to communication costs (i.e. $K$ is large). This may be the case with fast processors distributed across different machines, or in the setting of federated learning, where mobile devices collaborate to train a shared model while keeping their respective training datasets local [18].

This is captured by a graph consisting of $M$ parallel chains of length $TK$, with cross connections between the chains every $K$ nodes. Indexing the nodes as $v_{t,m,k}$, the nodes $v_{t,m,1} \rightarrow \cdots \rightarrow v_{t,m,K}$ form a chain, and $v_{t,m,K}$ is connected to $v_{t+1,m',1}$ for all $m' = 1..M$. This graph generalizes the layer graph by allowing $K$ sequential oracle queries between each complete synchronization; $K = 1$ recovers the layer graph, and the depiction above corresponds to $K = M = 3$. We refer to the computation between each synchronization step as a (communication) round.

The depth of this graph is $D = TK$ and the size is $N = TKM$. Focusing on the stochastic gradient oracle (the situation is similar for the prox oracle, except with the potential of smoothing a non-smooth objective, as discussed in Section 4.2), Theorem 1 yields the lower bound:

$$\Omega \left( \min \left\{ \frac{L}{\sqrt{TK}}, \frac{H}{T^2 K^2} \right\} + \frac{L}{\sqrt{TKM}} \right). \tag{16}$$

A natural algorithm for this graph is parallel SGD, where we run an SGD chain on each machine and average iterates during communication rounds, e.g. [18]. The updates are then given by:

$$x_{t,m,0} = \frac{1}{M} \sum_{m'} x_{t,m',K}$$
$$x_{t,m,k} = x_{t,m,k-1} - \eta_t \nabla f(x_{t,m,k-1}; z_{t,m,k}), \ k = 1, \ldots, K \tag{17}$$

(note that $x_{t,m,0}$ does not correspond to any node in the graph, and is included for convenience of presentation). Unfortunately, we are not aware of any satisfying analysis of such a parallel SGD approach. Instead, we consider two other algorithms in an attempt to match the lower bound (16). First, we can combine all $KM$ oracle accesses between communication rounds in order to form a single mini-batch, giving up on the possibility of sequential computation along the "local" $K$ node sub-paths. Using all $KM$ nodes to obtain stochastic gradient estimates at the same point, we can perform $T$ iterations of A-MB-SGD with a mini-batch size of $KM$, yielding an upper bound of

$$O \left( \frac{H}{T^2} + \frac{L}{\sqrt{TKM}} \right). \tag{18}$$

This is a reasonable and common approach, and it is optimal (up to constant factors) when $KM = O(\frac{L^2}{H^2} T^3)$ so that the statistical term is limiting. However, comparing (18) to the lower bound (16) we see a gap by a factor of $K^2$ in the optimization term, indicating the possibility for significant gains when $K$ is large (i.e. when we can process a large number of examples on each machine at each round). Improving the optimization term by this $K^2$ factor would allow statistical optimality as long as $M = O(T^3 K^3)$—-this is a very significant difference. In many scenarios we would expect a modest number of machines, but the amount of data on each machine could easily be much more than the number of communication rounds, especially if communication is across a wide area network.

In fact, when $K$ is large, a different approach is preferable: we can ignore all but a single chain and simply execute $KT$ iterations of sequential SGD, offering an upper bound of

$$O \left( \frac{L}{\sqrt{TK}} \right). \tag{19}$$

Although this approach seems extremely wasteful, it actually yields a better guarantee than (18) when $K \geq \tilde{\Omega}(T^3 L^2 / H)$. This is a realistic regime, e.g. in federated learning when computation is distributed across devices, communication is limited and sporadic and so only a relatively small number of rounds $T$ are possible, but each device already possesses a large amount of data. Furthermore, for non-smooth functions, (19) matches the lower bound (16).

Our upper bound on the complexity is therefore obtained by selecting either A-MB-SGD or single-machine sequential SGD, yielding a combined upper bound of

$$O\left(\min\left\{\frac{L}{\sqrt{TK}}, \frac{H}{T^2}\right\} + \frac{L}{\sqrt{TKM}}\cdot\right) \tag{20}$$

For smooth functions, there is still a significant gap between this upper bound and the lower bound (16). Furthermore, this upper bound is not achieved by a single algorithm, but rather a combination of two separate algorithms, covering two different regimes. This raises the question of whether there is a single, natural algorithm, perhaps an accelerated variant of the parallel SGD updates (17), that at the very least matches (20), and preferably also improves over them in the intermediate regime or even matches the lower bound (16).

**Active querying and SVRG**   All methods discussed so far used fully stochastic oracles, requesting a gradient (or prox computation) with respect to an independently and randomly drawn $z \sim \mathcal{P}$. We now turn to methods that also make active queries, i.e. draw samples from $\mathcal{P}$ and then repeatedly query the oracle, at different points $x$, but on the same samples $z$. Recall that all of our lower bounds are valid also in this setting.

With an active query gradient oracle, we can implement SVRG [14, 16] on an intermittent communication graph. More specifically, for an appropriate choice of $n$ and $\lambda$, we apply SVRG to the regularized empirical objective $\hat{F}_\lambda(x) = \frac{1}{n}\sum_{i=1}^{n} f(x; z_i) + \frac{\lambda}{2}\|x\|^2$

---

**Algorithm 1** SVRG

Parameters: $n, S, I$,    Sample $z_1, \ldots, z_n \sim \mathcal{P}$,    Initialize $x_0 = 0$
**for** $s = 1, 2, \ldots, S = \left\lfloor T / \left(\left\lceil \frac{n}{KM} \right\rceil + \left\lceil \frac{I}{K} \right\rceil\right)\right\rfloor$ **do**
    $\tilde{x} = x_{s-1}, \quad x_s^0 = \tilde{x}$
    $\tilde{g} = \nabla \hat{F}_\lambda(\tilde{x}) = \frac{1}{n}\sum_{i=1}^{n} \nabla f(\tilde{x}; z_i) + \lambda\tilde{x}$             $(*)$
    **for** $i = 1, 2, \ldots, I = \frac{H}{\lambda}$ **do**
        Sample $j \sim \text{Uniform}\{1, \ldots, n\}$
        $x_s^i = x_s^{i-1} - \eta\left(\left(\nabla f(x_s^{i-1}; z_j) + \lambda x_s^{i-1}\right) - \left(\nabla f(\tilde{x}; z_j) + \lambda\tilde{x}\right) + \tilde{g}\right)$     $(**)$
    **end for**
    $x_s = x_s^i$ for $i \sim \text{Uniform}\{1, \ldots, I\}$
**end for**
**Return** $x_S$

---

To do so, we first pick a sample $\{z_1, \ldots z_n\}$ (without actually querying the oracle). As indicated by Algorithm 1, we then alternate between computing full gradients on $\{z_1, \ldots z_n\}$ in parallel $(*)$, and sequential variance-reduced stochastic gradient updates in between $(**)$. The full gradient $\tilde{g}$ is computed using $n$ active queries to the gradient oracle. Since all of these oracle accesses are made at the same point $\tilde{x}$, this can be fully parallelized across the $M$ parallel chains of length $K$ thus requiring $n/KM$ rounds. The sequential variance-reduced stochastic gradient updates *cannot* be parallelized in this way, and must be performed using queries to the gradient oracle in just one of the $M$ available parallel chains, requiring $I/K$ rounds of synchronization. Consequently, each outer iteration of SVRG requires $\left\lceil \frac{n}{KM} \right\rceil + \left\lceil \frac{I}{K} \right\rceil$ rounds. We analyze this method using $\lambda = \Theta\left(\frac{L}{\sqrt{n}}\right)$, $I = \Theta\left(\frac{H}{\lambda}\right) = \Theta\left(\frac{H\sqrt{n}}{L}\right)$, and $n = \min\left\{\Theta\left(\frac{K^2 T^2 L^2}{H^2 \log^2(MKT/L)}\right), \Theta\left(\frac{MKT}{\log(MKT/L)}\right)\right\}$. Using the analysis of Johnson and Zhang [14], SVRG guarantees that, with an appropriate stepsize, we have $\hat{F}_\lambda(x_S) - \min_x \hat{F}_\lambda(x) \leq 2^{-S}$; the value of $x_S$ on the empirical objective also generalizes to the population, so $\mathbb{E}[f(x_S; z)] - \min_x \mathbb{E}[f(x; z)] \leq 2^{-S} + O\left(\frac{L}{\sqrt{n}}\right)$ (see [23]). With our choice of parameters, this implies upper bound (see Appendix E.3)

$$O\left(\left(\frac{H}{TK} + \frac{L}{\sqrt{TKM}}\right)\log\left(\frac{TKM}{L}\right)\right). \tag{21}$$

These guarantees improve over sequential SGD (17) as soon as $M > \log^2(TKM/L)$ and $K > H^2/L^2$, i.e. $L/\sqrt{TK} < L^2/H$. This is a very wide regime: we require only a moderate number of machines, and the second condition will typically hold for a smooth loss. Intuitively, SVRG does roughly the same number (up to a factor of two) of sequential updates as in the sequential SGD approach but it uses better, variance reduced updates. The price we pay is in the smaller total sample size since we keep calling the oracle on the same samples. Nevertheless, since SVRG only needs to calculate the "batch" gradient a logarithmic number of times, this incurs only an additional logarithmic factor.

Comparing (18) and (21), we see that SVRG also improves over A-MB-SGD as soon as $K > T\log(TKM/L)$, that is if the number of points we are processing on each machine each round is slightly more then the total number of rounds, which is also a realistic scenario.

To summarize, the best known upper bound for optimizing with intermittent communication using a pure stochastic oracle is (20), which combines two different algorithms. However, with active oracle accesses, SVRG is also possible and the upper bound becomes:

$$O\left(\min\left\{\frac{L}{\sqrt{TK}}, \left(\frac{H}{TK} + \frac{L}{\sqrt{TKM}}\right)\log\left(\frac{TKM}{L}\right), \frac{H}{T^2} + \frac{L}{\sqrt{TKM}}\right\}\right) \qquad (22)$$

## 5  Summary

Our main contributions in this paper are: (1) presenting a precise formal oracle framework for studying parallel stochastic optimization; (2) establishing tight oracle lower bounds in this framework that can then be easily applied to particular instances of parallel optimization; and (3) using the framework to study specific settings, obtaining optimality guarantees, understanding where additional assumptions would be needed to break barriers, and, perhaps most importantly, identifying gaps in our understanding that highlight possibilities for algorithmic improvement. Specifically,

- For non-smooth objectives and a stochastic prox oracle, smoothing and acceleration can improve performance in the layer graph setting. It is not clear if there is a more direct algorithm with the same optimal performance, e.g. averaging the answers from the prox oracle.

- In the delay graph setting, delayed update SGD's guarantee is not optimal. We suggest an alternative optimal algorithm, but it would be interesting and beneficial to understand the true behavior of delayed update SGD and to improve it as necessary to attain optimality.

- With intermittent communication, we show how different methods are better in different regimes, but even combining these methods does not match our lower bound. This raises the question of whether our lower bound is achievable. Are current methods optimal? Is the true optimal complexity somewhere in between? Even finding a single method that matches the current best performance in all regimes would be a significant advance here.

- With intermittent communication, active queries allow us to obtain better performance in a certain regime. Can we match this performance using pure stochastic queries or is there a real gap between active and pure stochastic queries?

The investigation into optimizing over $\mathcal{F}_{L,H,B}$ in our framework indicates that there is no advantage to the prox oracle for optimizing (sufficiently) smooth functions. This raises the question of what additional assumptions might allow us to leverage the prox oracle, which is intuitively much stronger as it allows global access to $f(\cdot; z)$. One option is to assume a bound on the variance of the stochastic oracle i.e. $\mathbb{E}_z[\|\nabla f(x; z) - \nabla F(x)\|^2] \le \sigma^2$ which captures the notion that the functions $f(\cdot; z)$ are somehow related and not arbitrarily different. In particular, if each stochastic oracle access, in each node, is based on a sample of $b$ data points (thus, a prox operation optimizes a sub-problem of size $b$), we have that $\sigma^2 \le L^2/b$. Initial investigation into the complexity of optimizing over the restricted class $\mathcal{F}_{L,H,B,\sigma}$ (where we also require the above variance bound), reveals a significant theoretical advantage for the prox oracle over the gradient oracle, even for smooth functions. This is an example of how formalizing the optimization problem gives insight into additional assumptions, in this case low variance, that are necessary for realizing the benefits of a stronger oracle.

**Acknowledgements**

We would like to thank Ohad Shamir for helpful discussions. This work was partially funded by NSF-BSF award 1718970 ("Convex and Non-Convex Distributed Learning") and a Google Research Award. BW is supported by the NSF Graduate Research Fellowship under award 1754881. AS was supported by NSF awards IIS-1447700 and AF-1763786, as well as a Sloan Foundation research award.

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
