[Supplementary Material]

# A Main lower bound lemma

This analysis closely follows that of previous work, specifically the proof of Theorem 1 in [27] and the proof of Lemma 4 in [8]. There are slight differences in the problem setup between this work and that of previous papers, thus we include the following analysis for completeness and to ensure that all of our results can be verified. We do not claim any significant technical novelty within this section.

Let $V = \{v_1, \ldots, v_k\}$ be a uniformly random orthonormal set of vectors in $\mathbb{R}^m$. All of the probabilities referred to in Appendix A will be over the randomness in the selection of $V$. Let $X = \{x_1, x_2, \ldots, x_N\}$ be a set of vectors in $\mathbb{R}^m$ where $\|x_i\| \leq 1$ for all $i \leq N$. Let these vectors be organized into disjoint subsets $X_1 \cup X_2 \cup \cdots \cup X_k = X$. Furthermore, suppose that for each $t \leq k$, the set $X_t$ is a deterministic function $X_t = X_t(X_{<t}, V)$, so it can also be expressed as $X_t = X_t(V)$.

Let $S_t = X_{\leq t} \cup V_{\leq t}$, let $P_t$ be the projection operator onto the span of $S_t$ and let $P_t^\perp$ be the projection onto the orthogonal complement of the span of $S_t$. As in [8, 27], define

$$G_t = G_t(V) = \left[\!\left[ \forall x \in X_t \; \forall j \geq t \; \left| \left\langle \frac{P_{t-1}^\perp x}{\|P_{t-1}^\perp x\|}, v_j \right\rangle \right| \leq \alpha \right]\!\right] \tag{23}$$

Finally, suppose that for each $t$, $X_t$ is of the form:

$$X_t(V) = X_t\left(V_{<t}\mathbb{1}_{G_{<t}} + V\mathbb{1}_{\neg G_{<t}}\right) \tag{24}$$

i.e. conditioned on the event $G_{<t}$, it is a deterministic function of $V_{<t}$ only (and not $v_t, \ldots, v_k$). We say that $\mathbb{P}[G_{<1}] = 1$, so $X_1$ is always independent of $V$.

First, we connect the events $G_t$ to a more immediately useful condition

**Lemma 1.** *[cf. Lemma 9 [8], Lemma 1 [27]] For any $c$, $k$, $N$, $V$, and $\{X_k\}_{t=1}^k$, let $\alpha = \min\left\{\frac{1}{4N}, \frac{c}{2(1+\sqrt{2N})}\right\}$ then for each $t \leq k$*

$$G_{\leq t} \implies G'_{\leq t} := \left[\!\left[ \forall r \leq t, \; \forall x \in X_r, \; \forall j \geq t \; |\langle x, v_j \rangle| \leq \frac{c}{2} \right]\!\right]$$

The proof of Lemma 1 involves straightforward linear algebra, and we defer it to Appendix A.1. By Lemma 1, $G_{<t} \subseteq G'_{<t}$, therefore the property (24) is implied by

$$X_t(V) = X_t\left(V_{<t}\mathbb{1}_{G'_{<t}} + V\mathbb{1}_{\neg G'_{<t}}\right) \tag{25}$$

Now, we state the main result which allows us to prove our lower bounds:

**Lemma 2.** *[cf. Lemma 4 [8], Lemma 4 [27]] For any $k \geq 1$, $N \geq 1$, $c \in (0,1)$, and dimension*

$$m \geq k + N + \max\left\{32N^2, \; \frac{8(1+\sqrt{2N})^2}{c^2}\right\} \log\left(2k^2 N\right)$$

*if the sets $X_1, \ldots, X_k$ satisfy the property (25) then*

$$\mathbb{P}\left[\forall t \leq k \; \forall x \in X_t \; \forall j \geq t \; |\langle x, v_j \rangle| \leq \frac{c}{2}\right] \geq \frac{1}{2}$$

The proof of Lemma 2 relies upon the following, whose proof we defer to Appendix A.1.

**Lemma 3.** *[cf. Lemma 11 [8], Lemma 3 [27]] Let $R$ be any rotation operator, $R^\top R = I$, that preserves $S_{t-1}$, that is $Rw = R^\top w = w$ for any $w \in \text{Span}(S_{t-1})$. Then the following conditional densities are equal*

$$p\left(V_{\geq t} \mid G_{<t}, V_{<t}\right) = p\left(RV_{\geq t} \mid G_{<t}, V_{<t}\right)$$

*Proof of Lemma 2.* This closely follows the proof of Lemma 4 [8] and Lemma 4 [27], with small modifications to account for the different setting.

Set $\alpha = \min\left\{\frac{1}{4N}, \frac{c}{2(1+\sqrt{2N})}\right\}$. Then by Lemma 1, since $X_1, \ldots, X_k$ satisfy the property (25)

$$\mathbb{P}\left[\forall t \leq k \; \forall x \in X_t \; \forall j \geq t \; |\langle x, v_j \rangle| \leq \frac{c}{2}\right] \geq \mathbb{P}[G_{\leq k}] = \prod_{t \leq k} \mathbb{P}[G_t \mid G_{<t}] \tag{26}$$

Focus on a single term in this product,

$$\mathbb{P}\left[G_t \mid G_{<t}\right] = \mathbb{E}_{V_{<t}}\left[\mathbb{P}\left[G_t \mid G_{<t}, V_{<t}\right]\right] \geq \inf_{V_{<t}} \mathbb{P}\left[G_t \mid G_{<t}, V_{<t}\right] \tag{27}$$

For any particular $V_{<t}$,

$$\mathbb{P}\left[G_t \mid G_{<t}, V_{<t}\right] = \mathbb{P}\left[\forall x \in X_t \; \forall j \geq t \; \left|\left\langle \frac{P_{t-1}^{\perp}x}{\|P_{t-1}^{\perp}x\|}, v_j \right\rangle\right| \leq \alpha \;\middle|\; G_{<t}, V_{<t}\right] \tag{28}$$

$$\geq 1 - \sum_{x \in X_t(V_{<t})} \sum_{j=t}^{k} \mathbb{P}\left[\left|\left\langle \frac{P_{t-1}^{\perp}x}{\|P_{t-1}^{\perp}x\|}, v_j \right\rangle\right| > \alpha \;\middle|\; G_{<t}, V_{<t}\right] \tag{29}$$

$$\geq 1 - \sum_{x \in X_t(V_{<t})} \sum_{j=t}^{k} \mathbb{P}\left[\left|\left\langle \frac{P_{t-1}^{\perp}x}{\|P_{t-1}^{\perp}x\|}, \frac{P_{t-1}^{\perp}v_j}{\|P_{t-1}^{\perp}v_j\|} \right\rangle\right| > \alpha \;\middle|\; G_{<t}, V_{<t}\right] \tag{30}$$

Conditioned on $G_{<t}$ and $V_{<t}$, the set $X_t = X_t(V_{<t})$ is fixed, as is the set $S_{t-1}$ and therefore $P_{t-1}^{\perp}$, so the first term in the inner product is a fixed unit vector. By Lemma 3, the conditional density of $v_j \mid G_{<t}, V_{<t}$ is spherically symmetric within the span onto which $P_{t-1}^{\perp}$ projects. Therefore, $\frac{P_{t-1}^{\perp}v_j}{\|P_{t-1}^{\perp}v_j\|}$ is distributed uniformly on the unit sphere in $\mathrm{Span}\,(S_{t-1})^{\perp}$, which has dimension at least $m' := m - (t-1) - \sum_{r=1}^{t-1}|X_r| \geq m - k + 1 - N$.

The probability of a fixed vector and a uniform random vector on the unit sphere in $\mathbb{R}^{m'}$ having inner product more than $\alpha$ is proportional to the surface area of the "end caps" of the sphere lying above and below circles of radius $\sqrt{1-\alpha^2}$, which is strictly smaller than the surface area of a full sphere of radius $\sqrt{1-\alpha^2}$. Therefore, for a given $x, v_j$

$$\mathbb{P}\left[\left|\left\langle \frac{P_{t-1}^{\perp}x}{\|P_{t-1}^{\perp}x\|}, \frac{P_{t-1}^{\perp}v_j}{\|P_{t-1}^{\perp}v_j\|} \right\rangle\right| > \alpha \;\middle|\; G_{<t}, V_{<t}\right] < \frac{\mathrm{SurfaceArea}_{m'}(\sqrt{1-\alpha^2})}{\mathrm{SurfaceArea}_{m'}(1)} \tag{31}$$

$$= \left(\sqrt{1-\alpha^2}\right)^{m'-1} \tag{32}$$

$$\leq \exp\left(-\frac{(m'-1)\alpha^2}{2}\right) \tag{33}$$

where we used that $1 - x \leq \exp(-x)$. Finally, this holds for each $t$, $x \in X_t$, and $j \geq t$, so

$$\mathbb{P}\left[G_{\leq k}\right] \geq \prod_{t \leq k} \inf_{V_{<t}} \mathbb{P}\left[G_t \mid G_{<t}, V_{<t}\right] \tag{34}$$

$$\geq \left(1 - kN \exp\left(-\frac{(m-k-N)\alpha^2}{2}\right)\right)^k \tag{35}$$

$$\geq 1 - k^2 N \exp\left(-\frac{\alpha^2}{2} \max\left\{32N^2, \frac{8(1+\sqrt{2N})^2}{c^2}\right\} \log\left(\frac{1}{2k^2N}\right)\right) \tag{36}$$

$$= \frac{1}{2} \tag{37}$$

Where we used that $m \geq k + N + \max\left\{32N^2, \frac{8(1+\sqrt{2N})^2}{c^2}\right\} \log\left(2k^2N\right)$ for (36). For (37), recall that we chose $\alpha = \min\left\{\frac{1}{4N}, \frac{c}{2(1+\sqrt{2N})}\right\}$ so $\max\left\{32N^2, \frac{8(1+\sqrt{2N})^2}{c^2}\right\} = \frac{2}{\alpha^2}$. $\qquad\square$

## A.1 Proof of Lemmas 1 and 3

**Lemma 1.** *[cf. Lemma 9 [8], Lemma 1 [27]] For any $c$, $k$, $N$, $V$, and $\{X_k\}_{t=1}^{k}$, let $\alpha = \min\left\{\frac{1}{4N}, \frac{c}{2(1+\sqrt{2N})}\right\}$ then for each $t \leq k$*

$$G_{\leq t} \implies G'_{\leq t} := \left[\!\left[\forall r \leq t, \; \forall x \in X_r, \; \forall j \geq t \; |\langle x, v_j \rangle| \leq \frac{c}{2}\right]\!\right]$$

*Proof.* This closely follows the proof of Lemma 9 [8], with slight modification to account for the different problem setup.

For $t \le k$ assume $G_{\le t}$. For any $x \in X_t$ and $j \ge t$

$$|\langle x, v_j \rangle| \le \|x\| \left|\left\langle \frac{x}{\|x\|}, P_{t-1}v_j \right\rangle\right| + \|x\| \left|\left\langle \frac{x}{\|x\|}, P_{t-1}^\perp v_j \right\rangle\right| \tag{38}$$

$$\le \|P_{t-1}v_j\| + \left|\left\langle \frac{P_{t-1}^\perp x}{\|x\|}, v_j \right\rangle\right| \tag{39}$$

$$\le \|P_{t-1}v_j\| + \left|\left\langle \frac{P_{t-1}^\perp x}{\|P_{t-1}^\perp x\|}, v_j \right\rangle\right| \tag{40}$$

$$\le \|P_{t-1}v_j\| + \alpha \tag{41}$$

First, we decomposed $v_j$ into its $S_{t-1}$ and $S_{t-1}^\perp$ components and applied the triangle inequality. Next we used that $\|x\| \le 1$ and that the orthogonal projection operator $P_{t-1}^\perp$ is self-adjoint. Finally, we used that the projection operator is non-expansive and the definition of $G_t$.

Next, we prove by induction on $t$ that for all $t \le k$ and $j \ge t$, the event $G_{\le t}$ implies that $\|P_{t-1}v_j\|^2 \le 2\alpha^2 \sum_{r=1}^{t-1} |X_r|$. As a base case ($t = 1$), observe that, trivially, $\|P_{t-1}v_j\|^2 = \|0v_j\|^2 = 0$. For the inductive step, fix any $t \le k$ and $j \ge t$ and suppose that $G_{\le t'} \implies \|P_{t'-1}v_j'\|^2 \le 2\alpha^2 \sum_{r=1}^{t'-1} |X_r|$ for all $t' < t$ and $j' \ge t'$. Let $\hat{P}_t$ project onto $\text{Span}\,(S_t \cup X_{t+1})$ (this includes $X_{t+1}$ in contrast with $P_t$) and let $\hat{P}_t^\perp$ project onto the orthogonal subspace. Since $\text{Span}\,(X_1 \cup X_2 \cup \cdots \cup X_{t-1} \cup V_{\le t-1}) = S_{t-1}$,

$$\left\{ \frac{P_{r-1}^\perp x}{\|P_{r-1}^\perp x\|} : r \le t-1,\ x \in X_r \right\} \cup \left\{ \frac{\hat{P}_{r-1}^\perp v_r}{\|\hat{P}_{r-1}^\perp v_r\|} : r \le t-1 \right\} \tag{42}$$

is a (potentially over-complete) basis for $S_{t-1}$. Using the triangle inequality and $G_{<t}$, we can therefore expand

$$\|P_{t-1}v_j\|^2 = \sum_{r=1}^{t-1} \sum_{x \in X_r} \left\langle \frac{P_{r-1}^\perp x}{\|P_{r-1}^\perp x\|}, v_j \right\rangle^2 + \sum_{r=1}^{t-1} \left\langle \frac{\hat{P}_{r-1}^\perp v_r}{\|\hat{P}_{r-1}^\perp v_r\|}, v_j \right\rangle^2 \tag{43}$$

$$\le \alpha^2 \sum_{r=1}^{t-1} |X_r| + \sum_{r=1}^{t-1} \frac{1}{\|\hat{P}_{r-1}^\perp v_r\|^2} \left\langle \hat{P}_{r-1}^\perp v_r, v_j \right\rangle^2 \tag{44}$$

We must now bound the second term of (44). Focusing on the inner product in the numerator for one particular $r < t$:

$$\left|\left\langle \hat{P}_{r-1}^\perp v_r, v_j \right\rangle\right| = \left|\langle v_r, v_j \rangle - \left\langle \hat{P}_{r-1}v_r, v_j \right\rangle\right| \tag{45}$$

$$= \left|\left\langle \hat{P}_{r-1}v_r, v_j \right\rangle\right| \tag{46}$$

$$\le |\langle P_{r-1}v_r, v_j \rangle| + \sum_{x \in X_r} \left|\left\langle \frac{P_{r-1}^\perp x}{\|P_{r-1}^\perp x\|}, v_r \right\rangle \left\langle \frac{P_{r-1}^\perp x}{\|P_{r-1}^\perp x\|}, v_j \right\rangle\right| \tag{47}$$

$$\le \|P_{r-1}v_r\|\,\|P_{r-1}v_j\| + |X_r|\alpha^2 \tag{48}$$

$$\le 2\alpha^2 \sum_{i=1}^{r-1} |X_i| + |X_r|\alpha^2 \tag{49}$$

$$\le \frac{\alpha}{2} \tag{50}$$

First, we used that $\hat{P}_{r-1}^\perp = I - \hat{P}_{r-1}$, then that $v_r \perp v_j$. Next, we applied the definition of $\hat{P}_{r-1}$ and the triangle inequality. To get (48) we use the Cauchy-Schwarz inequality on the first term, and the definition of $G_r$ for the second. Finally, we use the inductive hypothesis and that $\alpha \le \frac{1}{4N}$.

We have now upper bounded the inner products in the second term of (44), and it remains to lower bound the norm in the denominator. We can rewrite

$$\left\| \hat{P}_{r-1}^\perp v_r \right\|^2 = \left\langle \hat{P}_{r-1}^\perp v_r,\, v_r \right\rangle \tag{51}$$

$$= \langle v_r,\, v_r \rangle - \left\langle \hat{P}_{r-1} v_r,\, v_r \right\rangle \tag{52}$$

$$\geq 1 - \langle P_{r-1} v_r,\, v_r \rangle - \sum_{x \in X_r} \left\langle \frac{P_{r-1}^\perp x}{\left\| P_{r-1}^\perp x \right\|},\, v_r \right\rangle^2 \tag{53}$$

$$\geq 1 - \left\| P_{r-1} v_r \right\|^2 - |X_r|\, \alpha^2 \tag{54}$$

$$\geq 1 - 2\alpha^2 \sum_{i=1}^{r-1} |X_i| - |X_r|\, \alpha^2 \tag{55}$$

$$\geq \frac{1}{2} \tag{56}$$

Here we again used $\hat{P}_{r-1}^\perp = I - \hat{P}_{r-1}$ followed by an (over)expansion of $\hat{P}_{r-1}$. The remaining steps follow from the inductive hypothesis and fact that $\alpha \leq \frac{1}{4N}$. Combining (56) with (50) and returning to (44), we have that

$$\| P_{t-1} v_j \|^2 \leq \alpha^2 \sum_{r=1}^{t-1} |X_r| + \sum_{r=1}^{t-1} \frac{1}{\left\| \hat{P}_{r-1}^\perp v_r \right\|^2} \left\langle \hat{P}_{r-1}^\perp v_r,\, v_j \right\rangle^2 \tag{57}$$

$$\leq \alpha^2 \sum_{r=1}^{t-1} |X_r| + \sum_{r=1}^{t-1} \alpha^2 \tag{58}$$

$$\leq 2\alpha^2 \sum_{r=1}^{t-1} |X_r| \tag{59}$$

Therefore, for each $t \leq k$ and $j \geq t$ an upper bound $\| P_{t-1} v_j \|^2 \leq 2\alpha^2 \sum_{r=1}^{t-1} |X_r|$. Returning now to (41), we have that for any $t \leq k$, $x \in X_t$, and $j \geq t$ the event $G_{\leq t}$ implies

$$| \langle x,\, v_j \rangle | \leq \| P_{t-1} v_j \| + \alpha \tag{60}$$

$$\leq \alpha \left( 1 + \sqrt{2 \sum_{r=1}^{t-1} |X_r|} \right) \tag{61}$$

$$\leq \frac{c}{2} \tag{62}$$

where we used that $\alpha \leq \frac{c}{2\left(1+\sqrt{2N}\right)}$ $\qquad\square$

**Lemma 3.** *[cf. Lemma 11 [8], Lemma 3 [27]] Let $R$ be any rotation operator, $R^\top R = I$, that preserves $S_{t-1}$, that is $Rw = R^\top w = w$ for any $w \in \text{Span}\,(S_{t-1})$. Then the following conditional densities are equal*

$$p\,(V_{\geq t} \mid G_{<t}, V_{<t}) = p\,(RV_{\geq t} \mid G_{<t}, V_{<t})$$

*Proof.* This closely follows the proof of Lemma 11 [8].

First, we apply Bayes' rule to each density and use the fact that $RV_{<t} = V_{<t}$:

$$p\,(V_{\geq t} \mid G_{<t}, V_{<t}) = \frac{\mathbb{P}\,[G_{<t} \mid V]\, p(V)}{\mathbb{P}\,[G_{<t} \mid V_{<t}]\, p(V_{<t})} \tag{63}$$

$$p\,(RV_{\geq t} \mid G_{<t}, V_{<t}) = \frac{\mathbb{P}\,[G_{<t} \mid RV]\, p(RV)}{\mathbb{P}\,[G_{<t} \mid V_{<t}]\, p(V_{<t})} \tag{64}$$

Since $V$ has a spherically symmetric marginal distribution, $p(V) = p(RV)$. Therefore, it only remains to show that $\mathbb{P}\,[G_{<t} \mid V] = \mathbb{P}\,[G_{<t} \mid RV]$. The event $G_{<t}$ is determined by $V$ or by $RV$, thus both probabilities are either 0 or 1, so it suffices to show $\mathbb{P}\,[G_{<t} \mid V] = 1 \iff \mathbb{P}\,[G_{<t} \mid RV] = 1$.

Assume first $\mathbb{P}\left[G_{<t} \mid V\right] = 1$. Then for each $r < t$, $x \in X_r$, and $j \geq r$ $\left| \left\langle \frac{P_{r-1}^{\perp} x}{\|P_{r-1}^{\perp} x\|}, v_j \right\rangle \right| \leq \alpha$, and each set $X_r$ is a deterministic function of $V_{<r}$. Also, observe that for any $x \in X_r$ and $j \geq r$,

$$\left| \left\langle \frac{P_{r-1}^{\perp} x}{\|P_{r-1}^{\perp} x\|}, Rv_j \right\rangle \right| = \left| \left\langle \frac{R^{\top} P_{r-1}^{\perp} x}{\|P_{r-1}^{\perp} x\|}, v_j \right\rangle \right| = \left| \left\langle \frac{P_{r-1}^{\perp} x}{\|P_{r-1}^{\perp} x\|}, v_j \right\rangle \right| \leq \alpha \tag{65}$$

where we used that $P_{r-1}^{\perp} x \in \text{Span}(S_r) \subseteq \text{Span}(S_{t-1})$ so $R^{\top} P_{r-1}^{\perp} x = P_{r-1}^{\perp} x$. Therefore, it suffices to show that the sequence $X_1(RV), ..., X_t(RV) = X_1(V), ..., X_t(V)$ when $\mathbb{P}\left[G_{<t} \mid V\right] = 1$. We prove this by induction.

For the base case, by definition $X_1(RV) = X_1 = X_1(V)$. For the inductive step, suppose now that $X_{r'}(RV) = X_{r'}(V)$ for each $r' < r$. This, plus the fact that $\mathbb{P}\left[G_{<t} \mid V\right] = 1 \implies \mathbb{P}\left[G_{<r} \mid V\right] = 1$ together imply that $\mathbb{P}\left[G_{<r} \mid RV\right] = 1$. Thus, $X_r(RV) = X_r(RV_{<r}) = X_r(V_{<r})$. Therefore, we conclude that $\mathbb{P}\left[G_{<t} \mid V\right] = 1 \implies \mathbb{P}\left[G_{<t} \mid RV\right] = 1$, the reverse implication can be proven with a similar argument. □

## B  Proof of Theorem 1

**Theorem 1.** *Let $L, B \in (0, \infty)$, $H \in [0, \infty]$, $N \geq D \geq 1$, let $\mathcal{G}$ be any oracle graph of depth $D$ and size $N$ and consider the optimization problem $(\mathcal{G}, \mathcal{O}_{grad}, \mathcal{F}_{L,H,B})$. For any randomized algorithm $\mathcal{A} = (R_1, \ldots, R_N, \hat{X})$, there exists a distribution $\mathcal{P}$ and a convex, $L$-Lipschitz, and $H$-smooth function $f$ on a $B$-bounded domain in $\mathbb{R}^m$ for $m = O\left(\max\left\{N^2, D^3 N\right\} \log(DN)\right)$ such that*

$$\mathbb{E}_{\substack{z \sim \mathcal{P} \\ \hat{X} \sim \mathcal{A}}}\left[f(\hat{X}; z)\right] - \min_x \mathbb{E}_{z \sim \mathcal{P}}\left[f(x; z)\right] \geq \Omega\left(\min\left\{\frac{LB}{\sqrt{D}}, \frac{HB^2}{D^2}\right\} + \frac{LB}{\sqrt{N}}\right)$$

*Proof.* Assume for now that $B = 1$, the lower bound can be established for other values of $B$ by scaling inputs to our construction. Let

$$\ell = \min\left\{L, \frac{H}{10(D+1)^{1.5}}\right\} \qquad \eta = 10(D+1)^{1.5}\ell \tag{66}$$

and consider the following $\ell$-Lipschitz function:

$$\tilde{f}(x) = \max_{1 \leq r \leq D+1} \ell v_r^{\top} x - \frac{5\ell^2(r-1)}{\eta} \tag{67}$$

where the vectors $v_1, \ldots, v_{D+1}$ are an orthonormal set drawn uniformly at random from the unit sphere in $\mathbb{R}^m$. We use the $\eta$-Moreau envelope [5] of this function in order to prove our lower bound:

$$f(x) = \inf_y \left\{\tilde{f}(y) + \frac{\eta}{2}\|y - x\|^2\right\} \tag{68}$$

The random draw of $V$ defines a distribution over functions $f$. We will lower bound the expected suboptimality of any *deterministic* optimization algorithm's output and apply Yao's minimax principle at the end of the proof.

This function has the following properties:

**Lemma 4.** *The function $f$ is convex, $\ell$-Lipschitz, and $\eta$-smooth, with $\ell \leq L$ and $\eta \leq H$.*

Furthermore, optimizing $f$ is equivalent to "finding" the vectors $v_1, \ldots, v_{D+1}$. In particular, until a point that has a substantial inner product with all of $v_1, \ldots, v_{D+1}$ is found, the algorithm will remain far from the minimum:

**Lemma 5.** *For any $H, L > 0$, $D \geq 1$, and orthonormal $v_1, ..., v_{D+1}$, for any $x$ with $\left|v_{D+1}^{\top} x\right| \leq \frac{\ell}{\eta}$*

$$f(x) - \min_{x: \|x\| \leq 1} f(x) \geq \min\left\{\frac{L}{2\sqrt{D+1}}, \frac{H}{20(D+1)^2}\right\}$$

The function also has the property that if $x$ has a small inner product with $v_t, \ldots, v_{D+1}$, then the gradient oracle will reveal little information about $f$ when queried at $x$:

**Lemma 6.** *For any $x$ with $|\langle x, v_r \rangle| \leq \frac{\ell}{\eta}$ for all $r \geq t$, both the function value $f(x)$ and gradient $\nabla f(x)$ can be calculated from $v_1, \ldots, v_t$ only.*

In Appendix A, we studied the situation where orthonormal $v_1, \ldots, v_{D+1}$ are chosen uniformly at random and a sequence of sets of vectors $X_1, \ldots, X_{D+1}$ are generated as

$$X_t(V) = X_t \left( V_{<t} \mathbb{1}_{G'_{<t}} + V \mathbb{1}_{\neg G'_{<t}} \right) \tag{69}$$

where

$$G'_{<t} = \left[\!\left[ \forall r < t, \; \forall x \in X_r, \; \forall j \geq r \; |\langle x, v_j \rangle| \leq \frac{c}{2} \right]\!\right] \tag{70}$$

Take $c = \frac{2\ell}{\eta}$ and consider the dependency graph. Let $X_1$ be the set of queries made in vertices at depth 1 in the graph (i.e. they have no parents). Let $X_2$ be the set of queries made in vertices at depth 2 in the graph (i.e. their parents correspond to the queries in $X_1$). Continue in this way for each $t \leq D$, and let $X_{D+1} = \{\hat{x}\}$ correpond to the algorithm's output, which is allowed to depend on all queries and oracle responses in the graph, and thus has depth $D + 1$.

Supposing $G'_{<t}$, for all queries $x \in X_1 \cup \cdots \cup X_{t-1}$ and for all $r \geq t - 1$ we have $|\langle x, v_r \rangle| \leq \frac{c}{2} = \frac{\ell}{\eta}$. Therefore, by Lemma 6 all of the function evaluations and gradients returned by the stochastic gradient oracle are calculable from $v_1, \ldots, v_{t-1}$ only. Therefore, all of the queries in $X_t$ are a deterministic function of $V_{<t}$ (since we are currently considering only deterministic optimization algorithms), so $X_t$ satisfies the required decomposition property (69). Finally, the queries are required to be in the domain of $f$, thus they will have norm bounded by 1.

Therefore, by Lemma 2, when the dimension

$$m \geq D + 1 + N + \max \left\{ 32N^2, \; 200 \left( D + 1 \right)^3 \left( 1 + \sqrt{2N} \right)^2 \right\} \log \left( 2(D+1)^2 N \right) \tag{71}$$

with probability $1/2$, all $x \in X_1 \cup \cdots \cup X_{t+1}$ including the algorithm's output $\hat{x}$ satsify $|\langle x, v_{D+1} \rangle| \leq \frac{\ell}{\eta}$ so by Lemma 5

$$f(\hat{x}) - \min_{x : \|x\| \leq 1} f(x) \geq \min \left\{ \frac{L}{2\sqrt{D+1}}, \; \frac{H}{20(D+1)^2} \right\} \tag{72}$$

Therefore, by Yao's minimax principle for any randomized algorithm $\mathcal{A}$

$$\max_V \mathbb{E}_{\hat{X} \sim \mathcal{A}} \left[ f(\hat{X}) \right] - \min_{x : \|x\| \leq 1} f(x) \geq \min_{\text{deterministic } \mathcal{A}} \mathbb{E}_V \left[ f(\hat{x}) \right] - \min_{x : \|x\| \leq 1} f(x)$$

$$\geq \min \left\{ \frac{L}{4\sqrt{D+1}}, \; \frac{H}{40(D+1)^2} \right\} \tag{73}$$

The statistical term $\frac{L}{8\sqrt{N}}$ follows from Lemma 10. $\qquad\square$

### B.1 Deferred proofs

**Lemma 4.** *The function $f$ is convex, $\ell$-Lipschitz, and $\eta$-smooth, with $\ell \leq L$ and $\eta \leq H$.*

*Proof.* Since $\tilde{f}$ is the maximum of $\ell$-Lipschitz affine functions, it is convex and $\ell$-Lipschitz. Furthermore, by Proposition 12.29 [5], $f$, the $\eta$-Moreau Envelope of $\tilde{f}$ is $\eta$-smooth and

$$\nabla f(x) = \eta \left( x - \arg \min_y \tilde{f}(y) + \frac{\eta}{2} \|y - x\|^2 \right) \tag{74}$$

The minimizing $y$ satisfies that $\eta(x - y) \in \partial \tilde{f}(y)$ (where $\partial \tilde{f}(y)$ denotes the set of subgradients of $\tilde{f}$ at $y$), and since $\tilde{f}$ is $\ell$-Lipschitz this implies that $\|\nabla f(x)\| \leq \ell$. $\qquad\square$

**Lemma 5.** *For any $H, L > 0$, $D \geq 1$, and orthonormal $v_1, ..., v_{D+1}$, for any $x$ with $\left| v_{D+1}^\top x \right| \leq \frac{\ell}{\eta}$*

$$f(x) - \min_{x : \|x\| \leq 1} f(x) \geq \min \left\{ \frac{L}{2\sqrt{D+1}}, \; \frac{H}{20(D+1)^2} \right\}$$

*Proof.* First

$$\min_{x:\|x\|\leq 1} f(x) \leq f\left(-\sum_{r=1}^{D+1} \frac{v_r}{\sqrt{D+1}}\right) \leq \tilde{f}\left(-\sum_{r=1}^{D+1} \frac{v_r}{\sqrt{D+1}}\right) \leq -\frac{\ell}{\sqrt{D+1}} \quad (75)$$

Now, for an arbitrary point $x$ such that $\left|v_{D+1}^\top x\right| \leq \frac{\ell}{\eta} = \frac{1}{10(D+1)^{1.5}}$, consider

$$y^* = \text{prox}_{\tilde{f}}(x,\eta) = \arg\min_y \left\{ \max_{1\leq r\leq D+1} \left(\ell v_r^\top y - \frac{5\ell^2(r-1)}{\eta}\right) + \frac{\eta}{2}\|y-x\|^2 \right\} \quad (76)$$

Since $y^*$ is the minimizer, $\eta(x-y^*) \in \partial\tilde{f}(y^*)$ and since $\tilde{f}$ is $\ell$-Lipschitz, $\|x-y^*\| \leq \frac{\ell}{\eta}$. Thus $v_{D+1}^\top y^* \geq -\frac{2\ell}{\eta}$ and

$$f(x) = \tilde{f}(y^*) + \frac{\eta}{2}\|y^*-x\|^2 \quad (77)$$

$$= \max_{1\leq r\leq D+1} \left(\ell v_r^\top y^* - \frac{5\ell^2(r-1)}{\eta}\right) + \frac{\eta}{2}\|y^*-x\|^2 \quad (78)$$

$$\geq \ell v_{D+1}^\top y^* - \frac{5\ell^2 D}{\eta} \quad (79)$$

$$\geq -\frac{2\ell^2}{\eta} - \frac{5\ell^2 D}{\eta} \quad (80)$$

$$\geq -\frac{5\ell^2(D+1)}{\eta} \quad (81)$$

Combining (75) and (81), for any $x$ such that $\left|v_{D+1}^\top x\right| \leq \frac{\ell}{\eta}$

$$f(x) - \min_{x:\|x\|\leq 1} f(x) \geq \frac{\ell}{\sqrt{D+1}} - \frac{5\ell^2(D+1)}{\eta} = \min\left\{\frac{L}{2\sqrt{D+1}}, \frac{H}{20(D+1)^2}\right\} \quad (82)$$

$\square$

**Lemma 6.** *For any $x$ with $|\langle x, v_r\rangle| \leq \frac{\ell}{\eta}$ for all $r \geq t$, both the function value $f(x)$ and gradient $\nabla f(x)$ can be calculated from $v_1,\ldots,v_t$ only.*

*Proof.* Let $x$ be a point such that $\left|v_r^\top x\right| \leq \frac{\ell}{\eta}$ for all $r \geq t$. By Proposition 12.29 [5]

$$\nabla f(x) = \eta\left(x - \text{prox}_{\tilde{f}}(x,\eta)\right) \quad (83)$$

Since $f$ is $\ell$-Lipschitz (Lemma 4), $\left\|x - \text{prox}_{\tilde{f}}(x,\eta)\right\| \leq \frac{\ell}{\eta}$. Consequently, for $y^* = \text{prox}_{\tilde{f}}(x,\eta)$ we have $\left|v_r^\top y^*\right| \leq \frac{2\ell}{\eta}$ for all $r \geq t$. Furthermore,

$$\nabla f(x) = \eta(x-y^*) \in \text{conv}\left\{\ell v_r : r \in \arg\max_{1\leq r\leq D+1}\left(\ell v_r^\top y^* - \frac{5\ell^2(r-1)}{\eta}\right)\right\} \quad (84)$$

For any $r > t$

$$\ell v_r^\top y^* - \frac{5\ell^2(r-1)}{\eta} \leq \frac{2\ell^2}{\eta} - \frac{5\ell^2(r-1)}{\eta} = -\frac{5\ell^2\left(r-\frac{7}{5}\right)}{\eta} \quad (85)$$

Whereas

$$\ell v_t^\top y^* - \frac{5\ell^2(t-1)}{\eta} \geq -\frac{2\ell^2}{\eta} - \frac{5\ell^2(t-1)}{\eta} = -\frac{2\ell^2\left(t-\frac{3}{5}\right)}{\eta} \quad (86)$$

For any $r > t$ (85) is less than (86), thus no $r > t$ can be in the $\arg\max$ in (84). Therefore, using only $v_1,\ldots,v_t$ we can calculate

$$f(x) = \inf_y \left\{ \max_{1\leq r\leq D+1}\left(\ell v_r^\top y - \frac{5\ell^2(r-1)}{\eta}\right) + \frac{\eta}{2}\|y-x\|^2 \right\} \quad (87)$$

$$= \inf_y \left\{ \max_{1\leq r\leq t}\left(\ell v_r^\top y - \frac{5\ell^2(r-1)}{\eta}\right) + \frac{\eta}{2}\|y-x\|^2 \right\} \quad (88)$$

$$\quad (89)$$

and

$$\mathrm{prox}_{\tilde{f}}(x, \eta) = \arg\min_{y} \left\{ \max_{1 \leq r \leq D+1} \left( \ell v_r^\top y - \frac{5\ell^2(r-1)}{\eta} \right) + \frac{\eta}{2} \|y - x\|^2 \right\} \tag{90}$$

$$= \arg\min_{y} \left\{ \max_{1 \leq r \leq t} \left( \ell v_r^\top y - \frac{5\ell^2(r-1)}{\eta} \right) + \frac{\eta}{2} \|y - x\|^2 \right\} \tag{91}$$

from which we get $\nabla f(x) = \eta(x - \mathrm{prox}_{\tilde{f}}(x, \eta))$. $\qquad\square$

## C  Proof of Theorem 2

**Theorem 2.** *Let $L, B \in (0, \infty)$, $H \in [0, \infty]$, $N \geq D \geq 1$, let $\mathcal{G}$ be any oracle graph of depth $D$ and size $N$ and consider the optimization problem $(\mathcal{G}, \mathcal{O}_{prox}, \mathcal{F}_{L,H,B})$. For any randomized algorithm $\mathcal{A} = (R_1, \ldots, R_N, \hat{X})$, there exists a distribution $\mathcal{P}$ and a convex, $L$-Lipschitz, and $H$-smooth function $f$ on a $B$-bounded domain in $\mathbb{R}^m$ for $m = O\left(\max\left\{N^2, D^3 N\right\} \log(DN)\right)$ such that*

$$\mathbb{E}_{\substack{z \sim \mathcal{P} \\ \hat{X} \sim \mathcal{A}}} \left[ f(\hat{X}; z) \right] - \min_x \mathbb{E}_{z \sim \mathcal{P}} \left[ f(x; z) \right] \geq \Omega\left( \min\left\{ \frac{LB}{D}, \frac{HB^2}{D^2} \right\} + \frac{LB}{\sqrt{N}} \right)$$

*Proof.* Without loss of generality, assume $B = 1$, the lower bound can be proven for other values of $B$ by scaling inputs to our construction by $1/B$. Let

$$\eta = \min\{H, 2LD\} \qquad \gamma = \frac{4L}{\eta\sqrt{2D}} \qquad a = 2c = \frac{1}{\sqrt{8D^3}} \tag{92}$$

Define the following scalar function

$$\phi_c(z) = \begin{cases} 0 & |z| \leq c \\ 2(|z| - c)^2 & c < |z| \leq 2c \\ z^2 - 2c^2 & 2c < |z| \leq \gamma \\ 2\gamma |z| - \gamma^2 - 2c^2 & |z| > \gamma \end{cases} \tag{93}$$

It is straightforward to confirm that $\phi_c$ is convex, $2\gamma$-Lipschitz continuous, and 4-smooth. Let $\mathcal{P}$ be the uniform distribution over $\{1, 2\}$. Let $v_1, v_2, \ldots, v_{2D}$ be a set of orthonormal vectors drawn uniformly at random and define

$$f(x; 1) = \frac{\eta}{8} \left( -2a v_1^\top x + \phi_c\left(v_{2D}^\top x\right) + \sum_{r=3,5,7,\ldots}^{2D-1} \phi_c\left(v_{r-1}^\top x - v_r^\top x\right) \right) \tag{94}$$

$$f(x; 2) = \frac{\eta}{8} \left( \sum_{r=2,4,6,\ldots}^{2D} \phi_c\left(v_{r-1}^\top x - v_r^\top x\right) \right) \tag{95}$$

$$F(x) = \mathbb{E}_{z \sim \mathcal{P}} \left[ f(x; z) \right] = \frac{1}{2} \left( f(x; 1) + f(x; 2) \right) \tag{96}$$

$$= \frac{\eta}{16} \left( -2a v_1^\top x + \phi_c\left(v_{2D}^\top x\right) + \sum_{r=2}^{2D} \phi_c\left(v_{r-1}^\top x - v_r^\top x\right) \right) \tag{97}$$

The random choice of $V$ determines a distribution over functions $f(\cdot; 1)$ and $f(\cdot; 2)$. We will lower bound the expectation (over $V$) of the suboptimality of any deterministic algorithm's output, and then apply Yao's minimax principle.

First, we show that the functions $f(\cdot; 1)$ and $:= f(\cdot; 2)$ are convex, $L$-Lipschitz, and $H$-smooth:

**Lemma 7.** *For any $H, L \geq 0$, $D \geq 1$, and orthonormal $v_1, ..., v_{2D}$, and with $\eta$, $\gamma$, $a$, and $c$ chosen as in (92), $f(\cdot; 1)$ and $f(\cdot; 2)$ are convex, $L$-Lipschitz, and $H$-smooth.*

Next, we show that optimizing $F$ is equivalent to "finding" a large number of the vectors $v_1, \ldots, v_{2D}$:

**Lemma 8.** *For any $H, L \geq 0$, $D \geq 1$, and orthonormal $v_1, ..., v_{2D}$, and with $\eta$, $\gamma$, $a$, and $c$ chosen as in (92), for any $x$ such that $\left|v_r^\top x\right| \leq \frac{c}{2}$ for all $r > D$*

$$F(x) - \min_{x:\|x\| \leq 1} F(x) \geq \min\left\{\frac{L}{32D}, \frac{H}{64D^2}\right\}$$

Next, we show that at any point $x$ such that $\left|v_r^\top x\right| \leq \frac{c}{2}$ for all $r \geq t$, the function value, gradient, and prox of $f(\cdot; 1)$ and $f(\cdot; 2)$ at $x$ are calculable using $v_1, \ldots, v_t$ only:

**Lemma 9.** *For any $x$ such that $\left|v_r^\top x\right| \leq \frac{c}{2}$ for all $r \geq t$, and any $\beta \geq 0$ the function values, gradients, and proxs $f(x; 1)$, $f(x; 2)$, $\nabla f(x; 1)$, $\nabla f(x; 2)$, $\text{prox}_{f(\cdot, 1)}(x, \beta)$, and $\text{prox}_{f(\cdot, 2)}(x, \beta)$ are calculable using $\beta, x, v_1, \ldots, v_t$ only.*

In Appendix A, we studied the situation where orthonormal $v_1, \ldots, v_{2D}$ are chosen uniformly at random and a sequence of sets of vectors $X_1, \ldots, X_{2D}$ are generated as

$$X_t(V) = X_t\left(V_{<t}\mathbb{1}_{G'_{<t}} + V\mathbb{1}_{\neg G'_{<t}}\right) \tag{98}$$

where

$$G'_{<t} = \left[\!\left[\forall r < t, \ \forall x \in X_r, \ \forall j \geq r \ |\langle x, v_j\rangle| \leq \frac{c}{2}\right]\!\right] \tag{99}$$

Consider the dependency graph, and let $X_1$ be the set of queries made in vertices at depth 1 in the graph (i.e. they have no parents). Let $X_2$ be the set of queries made in vertices at depth 2 in the graph (i.e. their parents correspond to the queries in $X_1$). Continue in this way for each $t \leq D$, and then let $X_{D+1} = \{\hat{x}\}$ correpond to the output of the optimization algorithm, which for now is deterministic.

Suppose $G'_{<t}$. Then for all of the queries $x \in X_1 \cup \cdots \cup X_{t-1}$ and for all $r \geq t - 1$ we have $|\langle x, v_r\rangle| \leq \frac{c}{2}$. Therefore, by Lemma 9 the function values, gradients, and proxs of $f(\cdot; 1)$ and $f(\cdot; 2)$ are calculable based only on the query points and $v_1, \ldots, v_{t-1}$. Therefore, all of the queries in $X_t$ are a deterministic function of $V_{<t}$ only so $X_t$ satisfies the required decomposition property (98). Finally, the queries are required to be in the domain of $f$, thus they will have norm bounded by $B$.

Therefore, by Lemma 2 for

$$m \geq 2D + N + \max\left\{32N^2, \ 128B^2D^3(1 + \sqrt{2N})^2\right\}\log\left(8D^2N\right) \tag{100}$$

with probability $1/2$ for every $x \in X_1 \cup \cdots \cup X_{D+1}$ which includes $\hat{x}$, $|\langle x, v_r\rangle| \leq \frac{c}{2}$ for $r > D$, so by Lemma 8

$$f(\hat{x}) - \min_{x:\|x\| \leq 1} f(x) \geq \min\left\{\frac{L}{32D}, \frac{H}{64D^2}\right\} \tag{101}$$

Therefore,

$$\min_{\text{deterministic } \mathcal{A}} \mathbb{E}_V\left[f(\hat{x}) - \min_{x:\|x\| \leq 1} f(x)\right] \geq \min\left\{\frac{L}{64D}, \frac{H}{128D^2}\right\} \tag{102}$$

so by Yao's minimax principle, for any randomized algorithm $\mathcal{A}$

$$\max_V \mathbb{E}_{\hat{X} \sim \mathcal{A}}\left[f(\hat{X}) - \min_{x:\|x\| \leq 1} f(x)\right] \geq \min\left\{\frac{L}{64D}, \frac{H}{128D^2}\right\} \tag{103}$$

The statistical term $\frac{LB}{8\sqrt{N}}$ follows from Lemma 10. $\qquad\square$

## C.1   Deferred proof

**Lemma 7.** *For any $H, L \geq 0$, $D \geq 1$, and orthonormal $v_1, ..., v_{2D}$, and with $\eta$, $\gamma$, $a$, and $c$ chosen as in (92), $f(\cdot; 1)$ and $f(\cdot; 2)$ are convex, L-Lipschitz, and H-smooth.*

*Proof.* The functions $f(\cdot; 1)$ and $f(\cdot; 2)$ are a sum of linear functions and $\phi_c$, which is convex; therefore both are convex. Also, the scalar function $\phi_c$ is $2\gamma$-Lipschitz, so

$$\|\nabla f(x; 1)\|^2 = \left\|\frac{\eta}{8}\left(-2av_1 + \phi_c'\left(v_{2D}^\top x\right)v_{2D} + \sum_{r=3,5,7,\ldots}^{2D-1}\phi_c'\left(v_{r-1}^\top x - v_r^\top x\right)(v_{r-1} - v_r)\right)\right\|^2 \tag{104}$$

$$\leq \frac{\eta^2\left(a^2 + (2D-1)\gamma^2\right)}{16} \leq \frac{2D\eta^2\gamma^2}{16} = L^2 \tag{105}$$

where we used that $a = \frac{1}{\sqrt{8D^3}} < \gamma = \frac{4L}{\eta\sqrt{2D}}$. Similarly,

$$\|\nabla f(x; 2)\|^2 = \left\|\frac{\eta}{8}\left(\sum_{r=2,4,6,\ldots}^{2D}\phi_c'\left(v_{r-1}^\top x - v_r^\top x\right)(v_{r-1} - v_r)\right)\right\|^2 \leq \frac{2D\eta^2\gamma^2}{16} = L^2 \tag{106}$$

Therefore, $f(\cdot; 1)$ and $f(\cdot; 2)$ are $L$-Lipschitz. Furthermore, since $\phi_c$ is 4-smooth,

$$\left|v_i^\top \nabla^2 f(x; 1)v_j\right| \leq \begin{cases} \frac{\eta}{2} & |i-j| \leq 1 \\ 0 & |i-j| > 1 \end{cases} \quad \text{and} \quad \left|v_i^\top \nabla^2 f(x; 2)v_j\right| \leq \begin{cases} \frac{\eta}{2} & |i-j| \leq 1 \\ 0 & |i-j| > 1 \end{cases} \tag{107}$$

therefore, the maximum eigenvalue of $\nabla^2 f(\cdot; 1)$ and $\nabla^2 f(\cdot; 2)$ is at most $\eta \leq H$. $\qquad\square$

**Lemma 8.** *For any $H, L \geq 0$, $D \geq 1$, and orthonormal $v_1, \ldots, v_{2D}$, and with $\eta, \gamma, a$, and $c$ chosen as in (92), for any $x$ such that $\left|v_r^\top x\right| \leq \frac{c}{2}$ for all $r > D$*

$$F(x) - \min_{x:\|x\|\leq 1} F(x) \geq \min\left\{\frac{L}{32D}, \frac{H}{64D^2}\right\}$$

*Proof.* First, we upper bound $\min_{x:\|x\|\leq 1} F(x)$. Recalling that $a = \frac{1}{\sqrt{8D^3}}$, define

$$x^* = a\sum_{r=1}^{2D}(2D + 1 - r)v_r \tag{108}$$

$$\|x^*\|^2 = \frac{1}{8D^3}\left(\frac{2D(2D+1)(4D+1)}{6}\right) \leq 1 \tag{109}$$

For this $x^*$, $v_{r-1}^\top x^* - v_r^\top x^* = v_{2D}^\top x^* = a$ and with our choice of parameters (92), $2c = a < \gamma$, so that $\phi_c'(a) = 2a$, thus

$$\nabla F(x^*) = \frac{\eta}{16}\left(-2av_1 + \phi_c'\left(v_{2D}^\top x^*\right)v_{2D} + \sum_{r=2}^{2D}\phi_c'\left(v_{r-1}^\top x^* - v_r^\top x^*\right)(v_{r-1} - v_r)\right) \tag{110}$$

thus,

$$\nabla F(x^*)^\top v_1 = -2a + \phi_c'(a) = 0 \tag{111}$$
$$\nabla F(x^*)^\top v_r = -\phi_c'(a) + \phi_c'(a) = 0 \qquad\qquad 2 \leq r \leq 2D - 1 \tag{112}$$
$$\nabla F(x^*)^\top v_{2D} = -\phi_c'(a) + \phi_c'(a) = 0 \tag{113}$$

Since $\|x^*\| \leq 1$ and $\nabla F(x^*) = 0$, we conclude

$$\min_{x:\|x\|\leq 1} F(x) = F(x^*) = \frac{\eta}{16}\left(-2Da^2 - 4Dc^2\right) = -\frac{\eta Da^2}{4} = -\frac{\eta}{32D^2} \tag{114}$$

Let $X_D = \left\{x : \|x\| \leq 1, \left|v_r^\top x\right| \leq \frac{c}{2} \forall r > D\right\}$. We will now lower bound

$$\min_{x \in X_D} F(x) = \min_{x:\|x\|\leq 1} F(x) \quad \text{s.t.} \quad \left|v_r^\top x\right| \leq \frac{c}{2} \quad \forall r > D \tag{115}$$

Introducing dual variables $\lambda_{D+1}, ..., \lambda_{2D} \geq 0$, solving (115) amounts to finding an $x \in X_D$ and a set of non-negative $\lambda$s such that $\nabla F(x) = -\sum_{r=D+1}^{2D} \lambda_r \, \text{sign}\left(v_r^\top x\right) v_r$ and such that $\lambda_r \left(v_r^\top x - \frac{c}{2}\right) = 0$ for each $r$. Let

$$x_D = \sum_{r=1}^{D+1} \left(a\left(D+1-r\right) + \frac{c}{2}\right) v_r, \quad \lambda_{D+1} = 2a, \quad \lambda_{D+2} = \cdots = \lambda_k = 0 \tag{116}$$

Since $a\left(D+1-r\right) + \frac{c}{2} < a\left(2D+1-r\right)$ for $r \leq D+1$ and $\|x^*\| \leq 1$ it follows that $\|x_D\| \leq 1$. Furthermore, since $v_{r-1}^\top x_D - v_r^\top x_D = a$ for $2 \leq r \leq D+1$ and $2c = a < \gamma$, the gradient

$$\nabla F(x_D)^\top v_1 = -2a + \phi_c'\left(a\right) = 0 \tag{117}$$

$$\nabla F(x_D)^\top v_r = -\phi_c'\left(a\right) + \phi_c'\left(a\right) = 0 \qquad\qquad 2 \leq r \leq D \tag{118}$$

$$\nabla F(x_D)^\top v_{D+1} = -\phi_c'\left(a\right) + \phi_c'\left(\frac{c}{2}\right) = -2a = -\lambda_{D+1} \tag{119}$$

$$\nabla F(x_D)^\top v_r = 0 = -\lambda_r \qquad\qquad D+2 \leq r \leq 2D \tag{120}$$

Therefore,

$$\min_{x \in X_D} F(x) = F(x_D) = \frac{\eta}{16}\left(-Da^2 - ac - 2Dc^2\right) = -\frac{\eta(3D+1)a^2}{32} = -\frac{\eta(3D+1)}{256D^3} \tag{121}$$

Combining (114) and (121) we have that

$$\min_{x \in X_D} F(x) - \min_{x : \|x\| \leq 1} F(x) = F(x_D) - F(x^*)$$

$$= \frac{\eta}{32D^2} - \frac{\eta(3D+1)}{256D^3} \geq \frac{\eta}{32D^2} - \frac{\eta}{64D^2} = \min\left\{\frac{L}{32D}, \frac{H}{64D^2}\right\} \tag{122}$$

$\square$

**Lemma 9.** *For any $x$ such that $\left|v_r^\top x\right| \leq \frac{c}{2}$ for all $r \geq t$, and any $\beta \geq 0$ the function values, gradients, and proxs $f(x;1)$, $f(x;2)$, $\nabla f(x;1)$, $\nabla f(x;2)$, $\text{prox}_{f(\cdot,1)}(x, \beta)$, and $\text{prox}_{f(\cdot,2)}(x, \beta)$ are calculable using $\beta, x, v_1, \ldots, v_t$ only.*

*Proof.* Suppose that $x$ is a point such that $\left|v_r^\top x\right| \leq \frac{c}{2}$ for all $r \geq t$, and $\beta \geq 0$. Therefore, $\phi_c\left(v_{r-1}^\top x - v_r^\top x\right) = 0$ for $r > t$ so

$$f(x;1) = \frac{\eta}{8}\left(-2av_1^\top x + \phi_c\left(v_{2D}^\top x\right) + \sum_{r=3,5,7,\ldots}^{2D-1} \phi_c\left(v_{r-1}^\top x - v_r^\top x\right)\right) \tag{123}$$

$$= \frac{\eta}{8}\left(-2av_1^\top x + \sum_{r=3,5,7,\ldots}^{t} \phi_c\left(v_{r-1}^\top x - v_r^\top x\right)\right) \tag{124}$$

$$f(x;2) = \frac{\eta}{8}\left(\sum_{r=2,4,6,\ldots}^{2D} \phi_c\left(v_{r-1}^\top x - v_r^\top x\right)\right) \tag{125}$$

$$= \frac{\eta}{8}\left(\sum_{r=2,4,6,\ldots}^{t} \phi_c\left(v_{r-1}^\top x - v_r^\top x\right)\right) \tag{126}$$

Thus both $f(x;1)$ and $f(x;2)$ can be calculated from $x, v_1, \ldots, v_t$ only. Similarly, $\phi'_c\left(v_{r-1}^\top x - v_r^\top x\right) = 0$ for $r > t$ so

$$\nabla f(x;1) = \frac{\eta}{8}\left(-2av_1 + \phi'_c\left(v_{2D}^\top x\right) v_{2D} + \sum_{r=3,5,7,\ldots}^{2D-1} \phi'_c\left(v_{r-1}^\top x - v_r^\top x\right)(v_{r-1} - v_r)\right) \quad (127)$$

$$= \frac{\eta}{8}\left(-2av_1 + \sum_{r=3,5,7,\ldots}^{t} \phi'_c\left(v_{r-1}^\top x - v_r^\top x\right)(v_{r-1} - v_r)\right) \quad (128)$$

$$\nabla f(x;2) = \frac{\eta}{8}\left(\sum_{r=2,4,6,\ldots}^{2D} \phi'_c\left(v_{r-1}^\top x - v_r^\top x\right)(v_{r-1} - v_r)\right) \quad (129)$$

$$= \frac{\eta}{8}\left(\sum_{r=2,4,6,\ldots}^{t} \phi'_c\left(v_{r-1}^\top x - v_r^\top x\right)(v_{r-1} - v_r)\right) \quad (130)$$

Thus $\nabla f(x;1)$ and $\nabla f(x;2)$ can also be calculated from $x, v_1, \ldots, v_t$ only.

Now, we consider the proxs at such a point $x$. Let $t' = t$ if $t$ is odd, and $t' = t - 1$ if $t$ is even. Let $P$ be the projection operator onto $S = \mathrm{Span}\left(v_1, \ldots, v_{t'}\right)$ and let $P^\perp$ be the projection onto the orthogonal subspace, $S^\perp$. Then, since $f(x;1) = f(Px;1) + f(P^\perp x;1)$, we can decompose the prox:

$$\mathrm{prox}_{f(\cdot;1)}(x,\beta)$$

$$= \arg\min_{y} f(y;1) + \frac{\beta}{2}\|y - x\|^2 \quad (131)$$

$$= \arg\min_{y_1 \in S, y_2 \in S^\perp} f(y_1;1) + f(y_2;1) + \frac{\beta}{2}\left(\|y_1 - Px\|^2 + \|y_2 - P^\perp x\|^2\right) \quad (132)$$

$$= \arg\min_{y_1 \in S} \frac{\eta}{8}\left(-2av_1^\top y_1 + \sum_{r=3,5,7,\ldots}^{t'} \phi_c\left(v_{r-1}^\top y_1 - v_r^\top y_1\right)\right) + \frac{\beta}{2}\|y_1 - Px\|^2 \quad (133)$$

$$+ \arg\min_{y_2 \in S^\perp} \frac{\eta}{8}\left(\phi_c\left(v_{2D}^\top y_2\right) + \sum_{r=t'+2,t'+4,\ldots}^{2D-1} \phi_c\left(v_{r-1}^\top y_2 - v_r^\top y_2\right)\right) + \frac{\beta}{2}\|y_2 - P^\perp x\|^2 \quad (134)$$

$$= P^\perp x + \arg\min_{y_1 \in S} \frac{\eta}{8}\left(-2av_1^\top y_1 + \sum_{r=3,5,7,\ldots}^{t'} \phi_c\left(v_{r-1}^\top y_1 - v_r^\top y_1\right)\right) + \frac{\beta}{2}\|y_1 - Px\|^2 \quad (135)$$

Where we used that $\left|v_r^\top P^\perp x\right| = \left|v_r^\top x\right| \leq \frac{c}{2}$ for all $r > t'$, so setting $y_2 = P^\perp x$ achieves the minimum since every term in the expression is zero and function is non-negative. The vector $P^\perp x$ is calculable from $x, v_1, \ldots, v_{t'} \subseteq x, v_1, \ldots, v_t$, and similarly the second term is a minimization depends only on $\beta, x, v_1, \ldots, v_{t'} \subseteq \beta, x, v_1, \ldots, v_t$. A nearly identical argument shows that $\mathrm{prox}_{f(\cdot;2)}(x,\beta)$ has the same property. $\qquad\square$

## D  Statistical term

**Lemma 10.** *For any $L, B > 0$, there exists a distribution $\mathcal{P}$, and an L-Lipschitz, 0-smooth function $f$ defined on $[-B, B]$ such that the output $\hat{x}$ of any potentially randomized optimization algorithm which accesses a stochastic gradient or prox oracle at most $N$ times satisfies*

$$\mathbb{E}_{\hat{X}\sim\mathcal{A}}\left[\mathbb{E}_{z\sim\mathcal{P}}\left[f(\hat{X};z)\right]\right] - \min_{|x|\leq B}\mathbb{E}_{z\sim\mathcal{P}}\left[f(x;z)\right] \geq \frac{LB}{8\sqrt{N}}$$

*Proof.* Let $\epsilon > 0$ and $p \sim \mathrm{Uniform}\{p_1, p_{-1}\}$ where $p_1 = \frac{1+\epsilon}{2}$ and $p_{-1} = \frac{1-\epsilon}{2}$. Define $\mathcal{P}_p$ as

$$\mathbb{P}_{\mathcal{P}_p}[Z = 1] = 1 - \mathbb{P}_{\mathcal{P}_p}[Z = -1] = p \quad (136)$$

Then, let $f(x; z) = zLx$, so $\mathbb{E}_{z \sim \mathcal{P}_p}[f(x; z)] = (2p - 1)Lx$. When $p = p_1$, $(2p - 1) > 0$ so the minimizer is $x = -B$ with value $-LB(2p - 1) = -LB\epsilon$, and when $p = p_{-1}$, $(2p - 1) < 0$ so the minimizer is $x = B$, also with value $-LB\epsilon$. Furthermore, if $p = p_1$ and $x \geq 0$ then it is at least $LB\epsilon$-suboptimal, and if $p = p_2$ and $x \leq 0$ then it is also at least $LB\epsilon$-suboptimal.

Now consider any deterministic optimization algorithm which accesses the gradient or prox oracle $N$ times. Each gradient or prox oracle response can be simulated using a single $z \sim \mathcal{P}_p$, so the algorithm's output is $\hat{x} = \hat{x}(z_1, \ldots, z_N) \in [-B, B]$. Consider

$$\mathbb{E}_{p \sim \text{Uniform}\{p_1, p_{-1}\}, z \sim \mathcal{P}_p}[(2p - 1)L\hat{x}(z_1, \ldots, z_N) \,|\, z_1, \ldots, z_N]$$
$$\geq LB\epsilon \mathbb{P}_{p \sim \text{Uniform}\{p_1, p_{-1}\}, z \sim \mathcal{P}_p}[\text{sign}(\hat{x}(z_1, \ldots, z_N)) \neq \text{sign}(2p - 1) \,|\, z_1, \ldots, z_N] \quad (137)$$

Furthermore, the Bayes optimal estimate $\hat{x}$ of $\text{sign}(2p - 1)$ is

$$\hat{x}(z_1, \ldots, z_N) = \begin{cases} 1 & \frac{1}{N} \sum_{i=1}^N z_i \geq 0 \\ -1 & \frac{1}{N} \sum_{i=1}^N z_i < 0 \end{cases} \quad (138)$$

so

$$\mathbb{P}_{p \sim \text{Uniform}\{p_1, p_{-1}\}, z \sim \mathcal{P}_p}\left[\text{sign}(\hat{X}(z_1, \ldots, z_N)) \neq \text{sign}(2p - 1) \,\Big|\, z_1, \ldots, z_N\right]$$

$$\geq \mathbb{P}_{p \sim \text{Uniform}\{p_1, p_{-1}\}, z \sim \mathcal{P}_p}\left[\left|\frac{1}{N} \sum_{i=1}^N z_i - (2p - 1)\right| \geq \epsilon\right] \quad (139)$$

$$= \mathbb{P}_{z \sim \mathcal{P}_{p_{-1}}}\left[\left|\frac{1}{N} \sum_{i=1}^N z_i - \epsilon\right| \geq \epsilon\right] \quad (140)$$

This simply requires lower bounding the tail of the $\text{Binomial}(N, \frac{1-\epsilon}{2})$ distribution. Using Theorem 2.1 in [24],

$$\mathbb{P}_{z \sim \mathcal{P}_{p_{-1}}}\left[\left|\frac{1}{N} \sum_{i=1}^N z_i - \epsilon\right| \geq \epsilon\right] \geq 1 - \Phi\left(\frac{\epsilon N}{\sqrt{N(1 + \epsilon)(1 - \epsilon)}}\right) = 1 - \Phi\left(\frac{\epsilon \sqrt{N}}{\sqrt{1 - \epsilon^2}}\right) \quad (141)$$

where $\Phi$ is the distribution function of the standard normal. Let $\epsilon = \frac{1}{2\sqrt{N}}$, then $\frac{\epsilon \sqrt{N}}{\sqrt{1-\epsilon^2}} < \frac{3}{5}$ and

$$\mathbb{P}_{z \sim \mathcal{P}_{p_{-1}}}\left[\left|\frac{1}{N} \sum_{i=1}^N z_i - \epsilon\right| \geq \epsilon\right] \geq 1 - \Phi\left(\frac{3}{5}\right) \geq \frac{1}{4} \quad (142)$$

Therefore, we conclude that

$$\mathbb{E}_{p \sim \text{Uniform}\{p_1, p_{-1}\}, z \sim \mathcal{P}_p}[(2p - 1)L\hat{x}(z_1, \ldots, z_N) \,|\, z_1, \ldots, z_N] \geq \frac{LB\epsilon}{4} = \frac{LB}{8\sqrt{N}} \quad (143)$$

Therefore, by Yao's minimax principle, for any randomized algorithm $\mathcal{A}$

$$\max_{p \in \{p_1, p_{-1}\}} \mathbb{E}_{\hat{X} \sim \mathcal{A}}\left[\mathbb{E}_{z \sim \mathcal{P}_p}[f(\hat{X}; z)] - \min_x \mathbb{E}_{z \sim \mathcal{P}_p}[f(x; z)]\right] \geq \frac{LB}{8\sqrt{N}} \quad (144)$$

$\square$

# E  Supplement to Section 4

## E.1  Smoothed accelerated mini-batch SGD

Smoothed accelerated mini-batch SGD is the composition of two ingredients. First, we approximate the non-smooth $f$ with a smooth surrogate, and then perform accelerated mini-batch SGD on the surrogate [9, 15]. In particular, we use the $\beta$-Moreau envelope $f^{(\beta)}$ of $f$:

$$f^{(\beta)}(x; z) = \inf_y f(y; z) + \frac{\beta}{2} \|y - x\|^2 \quad (145)$$

Since $f$ is $L$-Lipschitz, $f^{(\beta)}$ has the following properties (Proposition 12.29 [5]):

1. $f^{(\beta)}$ is $\beta$-smooth
2. $\nabla f^{(\beta)}(x;z) = \beta(x - \mathrm{prox}_{f(\cdot;z)}(x,\beta))$
3. $f^{(\beta)}(x;z) \le f(x;z) \le f^{(\beta)}(x;z) + \frac{L^2}{2\beta}$ for all $x$

We use the prox oracle to execute A-MB-SGD on the $L$-Lipschitz and $\beta$-smooth $f^{(\beta)}$, with updates

$$w_t = \alpha y_t + (1-\alpha)x_t \tag{146}$$

$$y_{t+1} = w_t - \frac{\eta}{M}\sum_{i=1}^{M}\beta\left(w_t - \mathrm{prox}_{f(\cdot;z_i)}(w_t,\beta)\right) \tag{147}$$

$$x_{t+1} = \alpha y_{t+1} + (1-\alpha)x_t \tag{148}$$

The A-MB-SGD algorithm will converge on $f^{(\beta)}$ at a rate (see [9, 15])

$$\mathbb{E}\left[f(x_T;z)\right] - \min_x \mathbb{E}\left[f(x;z)\right] = O\left(\min\left\{\frac{L}{\sqrt{T}}, \frac{\beta}{T^2}\right\} + \frac{L}{\sqrt{MT}}\right) \tag{149}$$

Choosing $\beta = \min\{LT, H\}$ the conclude

$$\mathbb{E}\left[f(x_T;z)\right] - \min_x \mathbb{E}\left[f(x;z)\right] \le \mathbb{E}\left[f^{(\beta)}(x_T;z)\right] + \frac{L}{2T} - \min_x \mathbb{E}\left[f^{(\beta)}(x;z)\right] \tag{150}$$

$$= O\left(\min\left\{\frac{L}{\sqrt{T}}, \frac{\min\{LT, H\}}{T^2}\right\} + \frac{L}{\sqrt{MT}} + \frac{L}{T}\right) \tag{151}$$

$$= O\left(\min\left\{\frac{L}{T}, \frac{H}{T^2}\right\} + \frac{L}{\sqrt{MT}}\right) \tag{152}$$

which matches the lower bound in Theorem 2.

### E.2 Wait-and-collect accelerated mini-batch SGD

---
**Algorithm 2** "Wait-and-collect" accelerated minibatch SGD

---
Initialize $\hat{x} = \tilde{x} = x_0 = 0$,, parameter $\alpha$.
**for** $t = 1, 2, \ldots, T$ **do**
  **if** $\mod(t, 2\tau+1) \le \tau$ **then**
    Query stochastic gradient at $\tilde{x}$.
    Update $x_t \leftarrow x_{t-1}, \tilde{g} = 0$.
  **else if** $\mod(t, 2\tau+1) > \tau$ **and** $\mod(t, 2\tau+1) \le 2\tau$ **then**
    Update $x_t \leftarrow x_{t-1}$.
    Receive noisy gradient $g_{t-1-\tau}$, update $\tilde{g} \leftarrow \tilde{g} + (1/\tau) * g_{t-1-\tau}$
  **else if** $\mod(t, 2\tau+1) = 0$ **then**
    Update $x_t \leftarrow \tilde{x} - \eta\tilde{g}$.
    Update $\hat{x} \leftarrow \alpha\hat{x} + (1-\alpha)x_t, \tilde{x} \leftarrow \alpha\hat{x} + (1-\alpha)x_t$.
  **end if**
**end for**

---

### E.3 Analysis of technical results in Section 4.4

**Applying SVRG under intermittent synchronization graph** To apply SVRG method to solve stochastic convex optimization problems under intermittent synchronization graph. We adopt the approach by [16, 26], first we sample $n$ instances $\{z_1, ..., z_n\}$ and solve a regularized empirical risk minimization problem based on $\{z_1, ..., z_n\}$:

$$\min_x \hat{F}_\lambda(x) := \frac{1}{n}\sum_{i=1}^{n}f(x;z_i) + \frac{\lambda}{2}\|x\|^2, \tag{153}$$

where $\lambda$ is the regularization parameter will specified later. We will apply SVRG algorithm on the intermittent synchronization graph to solve above empirical objective (153) to certain sub-optimality.

The SVRG method works in stages, at each stage, we first use $n/KM$ communication rounds to calculate the full gradient of (153) at a reference point $\tilde{x}$, and then using a single chain to perform stochastic gradient updates, equipped with $\nabla \hat{F}(\tilde{w})$ to reduce the variance. We choose $\lambda \asymp L/(\sqrt{n}B)$, which will makes the objective (153) to be at least $L/(\sqrt{n}B)$-strongly convex, thus the condition number of (153) will be bounded by $O(H/(L/(\sqrt{n}B))) = O(H\sqrt{n}B/L)$. The SVRG analysis [14] requires the number of stochastic gradient updates to be scales as the condition number, so here we will use $O(H\sqrt{n}B/(LK))$ communication rounds to perform the stochastic updates, since one chain within each communication round has length $K$. Let $\hat{x}^* = \arg\min_x \hat{F}_\lambda(x)$, and let $\hat{x}_s$ to be the iterate after running the SVRG algorithm for $s$-stages. By the standard results of SVRG (Theorem 1 in [14]), we have

$$\mathbb{E}\left[\hat{F}_\lambda(\hat{x}_s)\right] - \hat{F}_\lambda(\hat{x}^*) \leq \left(\frac{1}{2}\right)^s.$$

By standard estimation-optimization error decomposition (e.g. Section 4 in [23]), we have

$$\mathbb{E}\left[F(\hat{x}_s)\right] - F(x^*) \leq 2\mathbb{E}\left[\hat{F}_\lambda(\hat{x}_s) - \hat{F}_\lambda(\hat{x}^*)\right] + \frac{\lambda B^2}{2} + O\left(\frac{L^2}{\lambda n}\right)$$

$$\leq \left(\frac{1}{2}\right)^s + O\left(\frac{LB}{\sqrt{n}}\right) = O\left(\frac{LB}{\sqrt{n}}\right), \tag{154}$$

given $s \asymp \log(n/(LB))$. Thus to implement SVRG successfully, we need to choose $n$ such that the following two conditions are satisfied:

$$\frac{n}{KM} * s \leq T, \quad \text{and} \quad \frac{H\sqrt{n}B}{LK} * s \leq T.$$

Thus we know by choosing $n$ below will satisfy above condition:

$$n \asymp \min\left\{\frac{K^2 T^2 L^2}{H^2 B^2 \log^2(MKT/L)}, \frac{MKT}{\log(MKT/L)}\right\},$$

substitute the scale of $n$ to (154) we get

$$\mathbb{E}\left[F(\hat{x}_s)\right] - F(x^*) \leq O\left(\frac{HB^2}{KT}\log\left(\frac{MKT}{L}\right) + \frac{LB}{\sqrt{MKT}}\left(\log\left(\frac{MKT}{L}\right)\right)^{1/2}\right)$$

$$\leq O\left(\left(\frac{HB^2}{KT} + \frac{LB}{\sqrt{MKT}}\right)\log\left(\frac{MKT}{L}\right)\right),$$

and we obtain the desired result.