[Reviews · NeurIPS 2018]

Reviewer 1



This paper proposes an oracle model for parallel algorithms for stochastic convex programming. There are two main oracle models that are studied: In the first one, the algorithm has access to an stochastic unbiased estimator of the gradient for a given feasible point, and in the second one one can query an stochastic proximal oracle (that is, the proximal operator for a random instance f(.,z), where z is sampled from the target distribution). All oracle access are obtained from i.i.d. samples from the target distribution. Interestingly, the paper introduces sample access architectures that can reflect parallelism, delayed updates or intermittent communication, and it turns out the obtained lower bounds are optimal for most of these cases. However, in some situations the matching upper bound comes from an unnatural algorithms, so it is left as open problem whether the natural algorithms used in practice are optimal as well. There are several technical contributions of this work. The first one is the incorporation of active queries: this is a standard scenario in statistical learning, but I haven't seen it before in the stochastic optimization framework. The second one, is a simple trick to parameterize nonsmooth and smooth functions simultaneously. Finally, there are these new "unnatural" but provably optimal methods for the proposed oracle models. Overall, I believe this paper is not only strong, but provocative and indicative of future lines of research. For this reason, I strongly support the acceptation of this paper. Detailed comments: 1. The idea of using Moreau-Yoshida regularization is already present in some works of lower bounds (e.g., https://arxiv.org/abs/1307.5001). In this work it is also used a simultaneous parameterization of various degrees of smoothness. 2. There are a few typos in the appendix. I will point out some of them. In page 11, line 24, the probability is of the inner product being smaller than c/2 (that last term is missing). In page 12, line 34, the exponential is missing a negative sign. In page 13, line 36, the factorization of ||x|| seems wrong since it is missing the division by ||x|| (it works overall, but the steps are confusing). In page 17, line 78, the upper bound of (eta/2)||y^{\ast}-x||^2 \geq \ell^2/(2\eta) goes in the wrong direction.

Reviewer 2



The paper presents lower bounds for parallel computation of population risk minimization over a ball using a DAG computation model, aiming to capture both computation and communication over parallel processors. Along the line of a series of papers [3,8,9,15,26,27] trying to provide lower bounds inspired lower bound of Nemirofski [19] on the serial setting, the present paper assumes queries are submitted to either a gradient or a prox oracle. Such queries constitute nodes in a graph, and the decision of which query to submit, as well as what value to "output" at a given node as a solution to the problem, depend only queries (and responses) observed *on the predecessors" of the nodes in the DAG. The authors provide lower bounds for the two oracles in this setting, depending on the Lipschitz constant and smoothness of the objective, as well as on the depth and size of the DAG and the radius of the constraint set. Going beyond these bounds, the authors discuss corresponding upper bounds for different topologies corresponding to different communication patterns, linking them to either existing algorithm and already derived upper bounds, or improving/tightening the gap with new proposed algorithms when the latter is not possible (though a few cases remain open). The paper is timely, rigorous, and extremely well-written, and the results are of significant consequence. Though the analysis builds upon earlier work (mostly [3,9,26]), the authors amend several assumptions made in the prior art regarding the type of communication allowed. The assumptions considered here (that communication is restricted to sharing computational outcomes across the DAG) are both natural and more general than prior art. The authors also identify and elucidate aspects of their proof and function construction that differ from prior work in the end of Section 3 (thank you!). There is no related work section, but the paper is well positioned with respect to related work spread out throughout the text. Section 4 in particular does a very good job linking lower bounds to a menagerie of existing/proposed algorithms, their upper bounds, and respective gaps (if they exist). Though this section is lucid and readable, it would be good to be more precise and formal about how A-MB-SGD in [10,14], the algorithm in [11], or SVRG [13,15] indeed fall under the corresponding DAG models (and indeed resort to respective oracle queries) as the ones used in the paper. To deal with space constraints and maintain the readability of Section 4, this formal derivation can be added in the supplement. Nits: Line 194 on Page 6, 11→ (11) Line 203 on Page 6, j=1…\tau → j=1,...,\tau (missing commas) Line 241 on Page 7, approach Instead → approach. Instead Line 259 on Page 7, posses → possesses Line 321 on Page 8, first order → first-order

Reviewer 3



Summary: This paper studies lower bounds for optimization problems based on a notion of "computation graph," which is to say, how long one has to have computations depend on one another. The authors prove lower bounds on optimization that scale as 1 / \sqrt{D}, where D is this computation depth (or 1 / D^2 if the function is smooth), with variants depending on the type of oracle access the methods have. These results are more or less new, though I think it would be nice if the authors gave a bit more credit to researchers who developed many of the technical techniques in the results. Criticisms and prior work: Many of the difficult technical lemmas in the supplement are due to Woodwarth and Srebro [WS16] (https://arxiv.org/abs/1605.08003), which were corrected by Carmon et al. [CDHS17] (https://arxiv.org/abs/1710.11606). In particular, Lemma 1 in this paper is Lemma 9 of [CDHS17], and Lemma 2 is basically Lemma 4 of [CDHS17]. Lemma 3 is Lemma 11 of [CDHS17]. The inequalities involving Surface Area [lines 429--434] are on page 20 of [WS16]. This type of more or less technical copying is unseemly, and does not give appropriate credit to the work of [WS16] and [CDHS17]. I would rather not reject the paper for this, because I think there is some good content in the remainder of the paper, but it is inappropriate to reprove results, without citation, that due to others. The lower bound constructions are similar to those of Nesterov in his 2004 book as well, but probably worth keeping explicitly. Additional comments: Line 109: These are not really the tightest possible lower bounds in that sense. The function f == 0 has lower bound 0 for any function. As such this sentence seems daft. Line 208: This is not particularly surprising. An algorithm that waits and collects mini-batches will have reduced variance gradient estimates, which are well-known to yield better convergence [e.g. the work of Dekel, Gilad-Bachrach, Shamir, Xiao]. Line 299: The word crisp is the wrong word here. Sharp or tight. Line 327: This line makes no sense to me, because I did not see real discussion of variance bounds in the paper. POST FEEDBACK REVIEW I believe the authors when they say they'll give appropriate citations to the prior literature. Happy to have the paper in, assuming that they will do this.